# Clouds drive differences in future surface melt over the Antarctic ice shelves

Christoph Kittel [1,2], Charles Amory [2], Stefan Hofer [3], Cécile Agosta [4], Nicolas C. Jourdain [2], Ella Gilbert [5], Louis Le Toumelin [6], Étienne Vignon [7], Hubert Gallée [2], and Xavier Fettweis [1]

[1]Department of Geography, UR SPHERES, University of Liège, Belgium
[2]Univ. Grenoble Alpes/CNRS/IRD/G-INP, IGE, Grenoble, France
[3]Department of Geosciences, University of Oslo, Oslo, Norway
[4]Laboratoire des Sciences du Climat et de l'Environnement, LSCE-IPSL, CEA-CNRS-UVSQ, Université Paris-Saclay, Gif-sur-Yvette, France
[5]British Antarctic Survey, High Cross, Madingley Road, Cambridge CB3 0ET, UK
[6]Univ. Grenoble Alpes, Université de Toulouse, Météo-France, CNRS, CNRM, Centre d'Études de la Neige, Grenoble, France
[7]Laboratoire de Météorologie Dynamique/IPSL/Sorbone Université/École Polytechnique/CNRS, UMR 8539, Paris, France

**Correspondence:** Christoph Kittel (ckittel@uliege.be)

**Abstract.** Recent warm atmospheric conditions have damaged the ice shelves of the Antarctic Peninsula through surface melt and hydrofracturing, and could potentially initiate future collapse of other Antarctic ice shelves. However, model projections with similar greenhouse gas scenarios suggest large differences in cumulative 21st century surface melting. So far it remains unclear whether these differences are due to variations in warming rates in individual models, or whether local feedback mechanisms on the surface energy budget could also play a notable role. Here we use the polar-oriented regional climate model MAR to study the physical mechanisms that would control future surface melt over the Antarctic ice shelves in high-emission scenarios RCP8.5 and SSP5-8.5. We show that clouds enhance future surface melt by increasing the atmospheric emissivity and longwave radiation towards the surface. Furthermore, we highlight that differences in meltwater production for the same climate warming rate depend on cloud properties and particularly cloud phase. Clouds containing a larger amount of supercooled liquid water lead to stronger melt, subsequently favouring the absorption of solar radiation due to the snow-melt-albedo feedback. As liquid-containing clouds are projected to increase the melt spread associated with a given warming rate, they could be a major source of uncertainties in projections of the future Antarctic contribution to sea level rise.

## 1 Introduction

Clouds are key drivers of the surface energy budget (SEB) of snow and ice. They can have opposing effects by reflecting solar (shortwave) radiation towards space and by emitting trapped energy through thermal (longwave) radiation towards the surface. The net cloud radiative effect - the balance between these opposite contributions - is notably determined by the surface albedo (Bintanja and van den Broeke, 1996; Hofer et al., 2017), and cloud properties, i.e their temperature (Stephens, 1984), structure (Barrett et al., 2017; Gilbert et al., 2020), and water phase (ice or liquid) (Lachlan-Cope, 2010; Van Tricht et al., 2016; Hines

et al., 2019; Gilbert et al., 2020). The absorption and reflection properties of clouds depend on the cloud optical depth (COD), which is partly linked to their liquid water content (Stephens, 1984; Zhang et al., 1996). Liquid-containing clouds, including both liquid-only and mixed-phase clouds, have a stronger effect on the COD and therefore on the SEB than ice clouds (Bennartz et al., 2013; Gorodetskaya et al., 2015; Hofer et al., 2019).

Clouds currently warm the Antarctic Ice Sheet (AIS) surface (Pavolonis and Key, 2003; Van Den Broeke et al., 2006). While most of the solar downwelling radiation (SWD) in summer is reflected by the high-albedo snow surface, clouds act as another source of incoming energy in the infrared spectrum, which can heat and melt snow (Bintanja and van den Broeke, 1996; Van Den Broeke et al., 2006) similarly as over bright surfaces of the Greenland Ice Sheet (Van Tricht et al., 2016). Abundant liquid-containing clouds associated with warm and moist air advection are responsible for intense melt events due to enhanced downwelling longwave fluxes (LWD) (Nicolas et al., 2017; Scott et al., 2019; Wille et al., 2019; Ghiz et al., 2021). These liquid-containing clouds can also become a significant source of incoming energy in winter and trigger surface melt even outside of the usual summer melt season (Kuipers Munneke et al., 2018b; Wille et al., 2019).

However, quantifying the influence of clouds on the SEB remains challenging at high latitudes. This is particularly true over the AIS where observations are scarce and expensive to maintain (Bromwich et al., 2012; Boucher et al., 2013). From a modelling perspective, the higher equilibrium climate sensitivities in Earth System Models (ESMs) from the recent 6th phase of the Coupled Model Intercomparison Project (CMIP6) than in CMIP5 models, the earlier 5th phase (Zelinka et al., 2020; Wyser et al., 2020; Wang et al., 2021), partly result from stronger positive cloud feedbacks over the southern ocean. This might explain why CMIP6-based projections suggest stronger changes over the Antarctic Ice Sheet, and especially a higher increase in melt over the margins (Kittel et al., 2021). Note that both global climate models and Earth System Models are broadly referred to as ESMs hereafter without any distinction between several degrees of model sophistications.

Little is known about how cloud-related uncertainties and more generally SEB will influence the future climate and surface mass balance projections over the Antarctic ice shelves. Surface melt in Antarctica is currently predominantly limited to Antarctic ice shelves, especially over the Peninsula (Trusel et al., 2013; Van Wessem et al., 2018; Agosta et al., 2019). Surface melt can damage the ice shelves, potentially initiate their collapse (van den Broeke, 2005) and increase the Antarctic contribution to sea level rise (SLR) through a speed-up in glacier flow (Scambos et al., 2014) and associated increase in ice discharge to the ocean. While melt amounts can be determined from a temperature-based diagnostic (Trusel et al., 2015), projected melt changes can vary considerably even at the same rate of warming (Kittel et al., 2021), and can lead to significant uncertainties in hydrofracturing risk (Gilbert and Kittel, 2021).

The aim of this work is to understand the physical drivers of changes in the SEB that produce large differences in melt projections over the Antarctic ice shelves. As such, we force the regional climate model (RCM) "Modèle Atmosphérique Régional" (MAR, Gallée and Schayes, 1994) with four ESMs from the CMIP5 (ACCESS1.3 and NorESM1-M) and CMIP6 (CNRM-CM6-1, CESM2) database using the highest greenhouse gas concentration pathways (respectively RCP8.5 and SSP5-8.5). A description of MAR and the experiments is given in Sect. 2. Section 3 details the regional evolution of the SEB and its different components over the AIS, and provides analysis of the physical drivers behind differences between projections. Finally, our results are discussed and summarised in Sect. 4.

## 2 Methods

### 2.1 The regional atmospheric model MAR

The Modèle Atmosphérique Régional (MAR) is a hydrostatic regional climate model specifically developed for polar areas (Gallée and Schayes, 1994). MAR has often been used to study the present and future climates of both the Antarctic (Agosta et al., 2019; Kittel et al., 2021) and Greenland ice sheets (Fettweis et al., 2020; Hofer et al., 2020). In this study, we used MARv3.11 whose specific adaptation and setup for the AIS is given in Agosta et al. (2019) and Kittel et al. (2021). The model has been thoroughly evaluated over the AIS against near-surface observations from automatic weather stations (Datta et al., 2018; Mottram et al., 2021; Kittel et al., 2021; Amory et al., 2021; Hofer et al., 2021) including radiative fluxes (Le Toumelin et al., 2021; Kittel, 2021; Hofer et al., 2021), SMB measurements (Kittel et al., 2018; Agosta et al., 2019; Donat-Magnin et al., 2020; Mottram et al., 2021; Kittel et al., 2021), melt estimates derived from both satellites (Datta et al., 2018; Donat-Magnin et al., 2020) and weather stations (Kittel et al., 2021), and satellite cloud cover (Hofer et al., 2021). MAR underestimates summer SWD by -6.9 $\mathrm{W\,m^{-2}}$ and LWD throughout the year by -9.9 $\mathrm{W\,m^{-2}}$ (Kittel, 2021).

It is important to note that MAR compares well with recent melt estimates and near-surface temperature observations (Kittel et al., 2021). This suggests a satisfactory representation of the SEB likely due to compensating turbulent fluxes whose impacts on the future SEB and melt is difficult to assess. A first comparison with CloudSat-Calipso product (described in Van Tricht et al. (2016); Lenaerts et al. (2017)) suggests that MAR underestimates the liquid water path (LWP, Fig. S1) but overestimates the ice (taking into account both ice and snow) water path (IWP, Fig. S2) around Antarctica, which has also been reported by other studies over the Arctic (e.g., Mattingly et al., 2020). This underestimation deserves further analyses with a comparison accounting for the limitations of the satellite product. However, this bias affects all simulations in an equivalent way, and its influence is likely removed in comparisons between different downscallings of ESMs, all produced with the same model physics. This should not preclude an explanation of the physical drivers behind the projected spread in melt illustrated in previous studies using MAR (Gilbert and Kittel, 2021; Kittel et al., 2021) but should be kept in mind when discussing the plausibility of these projections.

The cloud microphysics module of MAR solves conservation equations for five water species (cloud droplets, ice crystal, snow particles, rain drops, and specific humidity; Gallée, 1995) and the number of ice crystals (Messager et al., 2004). The model takes into account the influence of these water species on cloud radiative properties (Gallée and Gorodetskaya, 2010) and energy budget of each atmospheric layer in the radiative scheme inherited from the ECMWF ERA-40 reanalyses (Morcrette, 2002). MAR uses a broadband scheme for the longwave and shortwave radiations that integrates the values over the entire range of the two spectra. The radiative scheme uses the ice crystal, water vapour and cloud droplet concentrations from each atmospheric layer to determine the cloud optical properties. The snow particle concentration is implicitly taken into account by being partially included in the ice crystal concentration of each layer. The contribution of snow is expressed as an additional concentration for ice crystal by assuming that the total ratio of snow and ice crystal is similar to the ratio of their effective radii, i.e only 30% of snow is added in the ice crystal concentration input in the radiative scheme (Gallée and Gorodetskaya, 2010). The effect of rain droplets on radiation is neglected especially since the fall velocity of rain droplets used in MAR

(see Emde and Kahlig (1989)) induces that most of them reach the surface within one time-step of the radiative scheme. For shortwave radiation, the scheme uses the microphysics properties defined by Slingo (1989) for water clouds and by Fu (1996) for ice clouds while water and ice cloud properties for longwave radiation are respectively based on parameterisations detailed in Lindner and Li (2000) and Fu et al. (1998).

### 2.1.1 Surface Energy Budget (SEB)

The surface module SISVAT (Soil Ice Snow Vegetation Atmosphere Transfer; De Ridder and Schayes, 1997; De Ridder, 1997; Gallée and Duynkerke, 1997; Gallée et al., 2001; Lefebre et al., 2003) represents the evolution of snow and ice layer properties, including their albedo whose computation is inherited from CROCUS (Brun et al., 1992). SISVAT also deals with energy and mass exchanges between the atmosphere and the surface. SISVAT explicitly resolves the energy budget of 30 layers of snow and ice following Gallée and Duynkerke (1997). In particular, the surface temperature evolution depends on the net shortwave (SWN), net longwave (LWN), sensible heat (SHF) and latent heat (LHF) fluxes, but also on snow melting, liquid water refreezing and thermal diffusion into layer(s) immediately below. The excess in energy is used to warm the snowpack or to melt the surface snow/ice if the surface temperature has reached 0°C. Liquid water resulting from melt or rain can percolate vertically and refreeze in the snowpack.

In this study, we have approximated the SEB (Eq. 1) as :

$$SEB = SWN + LWN + LHF + SHF. \tag{1}$$

with positive fluxes directed towards the surface.

We neglect snow thermal diffusion and liquid water refreezing energy as the focus of this study is on the atmospheric factors that contribute to surface melting. The snow thermal diffusion is also considered to be an order of magnitude smaller than other radiative and turbulent fluxes (Van As et al., 2005). Furthermore, the snow thermal diffusion does not contribute to surface melting as during melt conditions the surface layer at 0°C induces a downward heat flux toward colder underlying layers. The thin layers of snow at the surface cannot hold much liquid water, in contrast to the deeper and thicker layers of the snowpack into which liquid water percolates. Refreezing therefore has a much higher warming potential in the deeper layers and only weakly contributes to surface warming. Finally, note that although refreezing increases with the production of liquid water via rain and surface melt, the projected increase in runoff indicates a decrease in the capacity of the snowpack to absorb liquid water (Donat-Magnin et al., 2021; Kittel et al., 2021; Gilbert and Kittel, 2021) and then in the refreezing potential, especially for larger warming rates. This highlights the predominant effect of the radiative - mostly SWN and LWN - or turbulent - mostly LHF and SHF - fluxes and justifies the simplified SEB equation.

### 2.1.2 Forcing datasets and experiments

Large-scale conditions are prescribed every 6 hours at the MAR boundaries. The forcing fields include information about air temperature, specific humidity, zonal and meridional wind speed components, and at the surface, pressure, sea temperature,

and sea ice concentration. MAR is also nudged in the upper atmosphere by large-scale temperature and wind components to constrain its atmospheric circulation (Agosta et al., 2019).

Most of the projections of the Antarctic surface melt have been based on direct outputs of ESMs (e.g., Seroussi et al., 2020) from CMIP5, or derived from them using statistical regressions (e.g., Trusel et al., 2015), while more recent climate models from CMIP6 now project stronger warmings at both regional (Antarctic) and global scales. Although the plausibility of (very) high climate sensitivity in the CMIP6 ESMs remains actively debated (Bjordal et al., 2020; Meehl et al., 2020; Sherwood et al., 2020; Zhu et al., 2020), these ESMs enable the evaluation of the sensitivity of the AIS to high temperature increases over the 21st century. We selected models from both CMIP5 and CMIP6 using the highest emission scenario (i.e, RCP8.5 for CMIP5 models and SSP5-8.5 for CMIP6). These scenarios are equivalent in terms of radiative forcing (+8.5 W m$^{-2}$) in 2100 (O'Neill et al., 2016). The detailed procedure that aims to select models that accurately represent the present Antarctic climate and maximise projected warming diversity can be found in Agosta et al. (2015), Barthel et al. (2020), and Kittel et al. (2021). In this study, MAR is forced by two CMIP5 models (ACCESS1.3 and NorESM-1-M) and two CMIP6 models (CNRM-CM6-1 and CESM2). These ESMs represent a large range of projected Antarctic warmings in 2100 qualified from weak (+3.2°C) to strong (+8.5°C) compared to the reference climate of 1981–2010. We performed our projections with MAR using a 35km spatial resolution over 1975–2100, discarding the six first years considered as spinup time. The evaluation of these MAR experiments can be found in Kittel et al. (2021).

The reference (present) period for computing the anomalies (hereafter referred to as changes) in this study is taken as the summer (December-January-February, DJF) average from 1981 to 2010 for MAR over ice shelves (melt, SEB components, cloud amount and properties, surface albedo). In the same way, we define the ESM warming as the mean changes in the summer (DJF) near-surface temperatures over the Antarctic region, i.e 90°S–60°S (near-surface warming) compared to 1981-2010. Since more than 80% of the local annual melt still occurs in summer by 2100 (excepted over the Peninsula where it is more than 50%), we only discuss the summer changes.

## 3 Results

### 3.1 Contributions to summer melt increase

Our four simulations project an increase in cumulative summer melt over the ice shelves that strongly differs depending on the forcing ESM during the 21st century (Fig. 1). We find a factor of ∼3.9 between the lowest and highest cumulative melt changes over the 21st century, despite equivalent radiative forcing from greenhouse gases. MAR driven by NorESM1-M simulates a cumulative melt increase of ∼7600 Gt during the 21st century (i.e the lowest melt increase), while the increase reaches ∼30150 Gt when MAR is driven by CNRM-CM6-1 (i.e the highest melt projection). This spread in projected melt (despite an equivalent concentration pathway) is as large as differences in multimodel estimates of Antarctic ice shelf surface melt between low- and high-concentration pathways by 2100 (Trusel et al., 2015; Kittel et al., 2021).

Similarly, our MAR experiments project different melt increases over each region depending on the forcing ESM. Between the lowest and the highest increases, we found a factor of ∼2.5 over the Antarctic Peninsula (AP) (Fig. 1f), ∼4.4 over the

East Antarctic Ice Shelves (EAIS) (Fig. 1k), and a factor of ∼5 over the West Antarctic Ice Shelves (WAIS) where we also included Ross and Ronne-Filchner ice shelves (Fig. 1p). While the NorESM1-M and the ACCESS1.3 experiments project different increases over each region, the CNRM-CM6-1 and CESM2 experiments mostly differ over the WAIS. There is indeed a factor ∼1.6 between these two projections over the WAIS despite a similar ESM warming. The WAIS (with Ross and Ronne-Filchner) appears to be a region of major uncertainties as the differences in that specific sector dominate the Antarctic signal. Before discussing the SEB drivers leading to large differences in surface melt increase over the WAIS, we will first analyse the two other sectors (EAIS and AP) because the changes in the SEB (and associated processes) are different in each region.

Over the AP, all flux changes are projected to positively contribute to the melt increases. MAR projects an similar positive contribution of radiative fluxes (LWN and SWN) for each experiment except when forced by CESM2 where the increase in SWN is stronger than in LWN. The relatively lower increase in LWN in this experiment results from the competitive effect of more opaque clouds (higher optical depth), but significant decreased cloud cover over the AP (Fig. S3). These changes in cloud cover also contribute to decrease snow precipitation (Kittel et al., 2021). The combination of increased melt and reduced snowfall leads to a large decrease in the albedo (Fig. S4), explaining the higher contribution of SWN in the CESM2 experiment (Fig. 1p). It is interesting to note that the positive contribution of both sensible and latent turbulent fluxes is specific to the ice shelves of the AP. Recent studies (Kuipers Munneke et al., 2012; van Wessem et al., 2014; Kuipers Munneke et al., 2018a; Datta et al., 2019) have suggested that warm air advections (notably during foehn events) are an important source of energy over the Peninsula producing strong melt over the present climate. MAR simulations project a strong local warming due to warmer and moister air advections inducing higher precipitation (Kittel et al., 2021) but also larger melt rates. Since the snow/ice-covered surface cannot warm higher than the melting temperature, warmer air advections also increase the thermal inversion near the surface and then increase SHF.

The melt increase over the EAIS is projected to be dominated by the increase in radiative fluxes and especially SWN. The NorESM1-M experiment excepted, all experiments project a stronger increase in SWN than LWN with a factor between ∼1.7 to ∼3.7. The large increase in SWN results from the decrease in albedo (Fig. S4) due to melt (and associated melt-albedo feedback) and not a reduction of high-albedo snowfall that is projected to increase over the ice shelves of this sector (Kittel et al., 2021). The melt-albedo feedback also explains the low contribution of SWN relative to LWN in the NorESM1-M experiment as melt is likely too weak to actually trigger it taking into account the increase in fresh snow.

The WAIS sector including the Ross and Ronne-Filchner ice shelves drives the Antarctic-scale differences in projected melt. Following all MAR projections, the radiative fluxes explain the increase in melt while turbulent fluxes have a negative contribution. However, only LWN is projected to strongly increase and explains uncertainties in melt. The SWN contributions of MAR forced by CNRM-CM6-1 and CESM2 (and to a lesser extent ACCESS1.3) are almost equivalent, whereas the CNRM-CM6-1 experiment projects a much larger (∼twice as large as) increase in LWN than all the other simulations. MAR projects an increase in cloud cover (Fig. S3) enhancing LWN but this is not sufficient to explain the projected differences (see hereafter). It is important to note that results in this sector are mostly driven by the Ross and Ronne-Filchner ice shelves due to their surface areas.

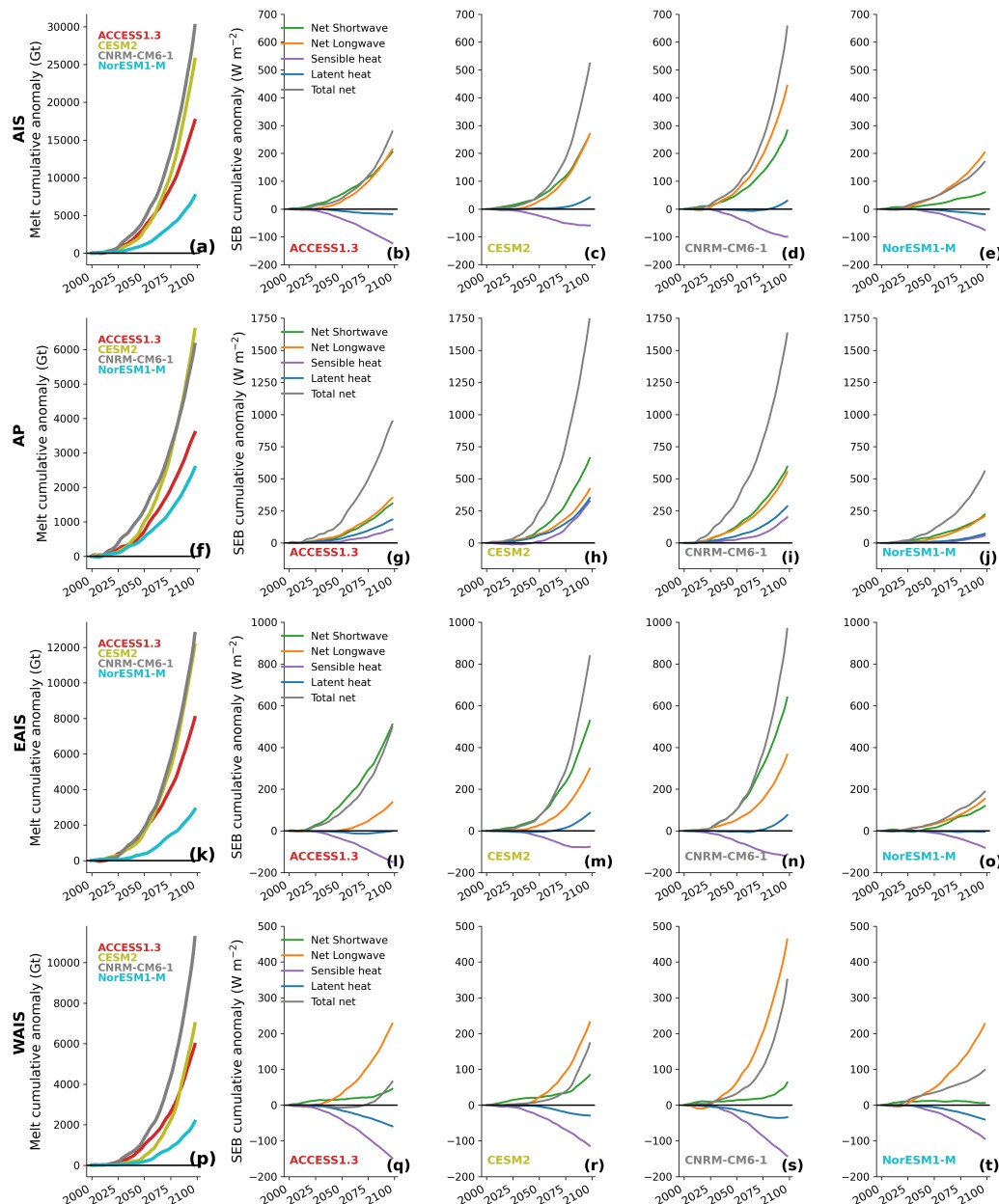

**Figure 1.** Cumulative surface melt (Gt) and SEB changes (W m$^{-2}$) over the Antarctic ice shelves. The first row (a-e) shows the cumulative integrated surface melt and averaged SEB components over the whole Antarctic ice shelves, while the second row (f,g,h,i,j) is for the Antarctic Peninsula, the third row for East Antarctic sector (k,l,m,n,o) and the fourth row (p,q,r,s,t) for the West Antarctic sector including Ross and Ronne-Filchner ice shelves. The second to the fifth columns represent the cumulative changes for each SEB component (green : net shortwave, orange : net longwave, purple : sensible heat, blue : latent heat) for each MAR simulation (the second row : forced by ACCESS1.3, the third row : CESM2, the fourth row : CNRM-CM6-1, the fifth row : NorESM1-M)

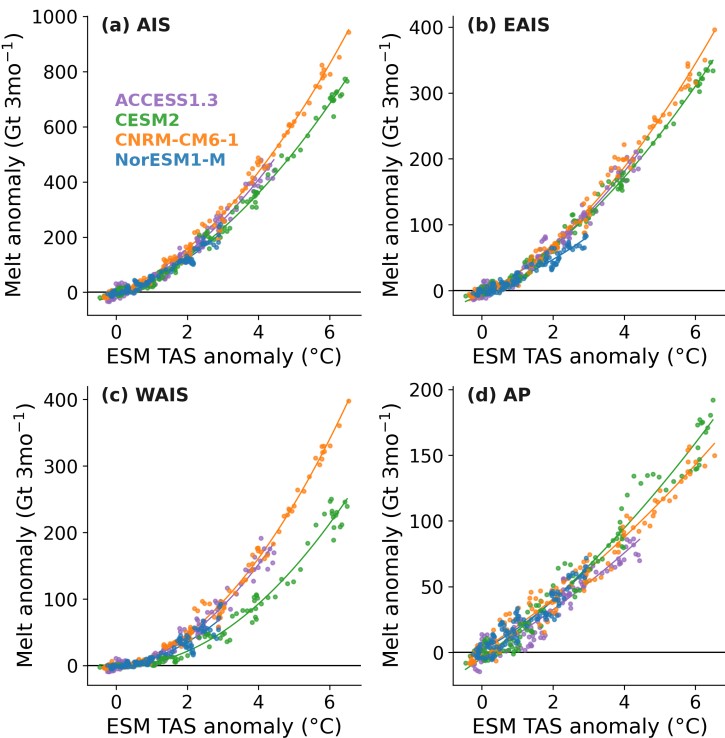

**Figure 2.** Mean summer melt changes ($\mathrm{Gt\,3mo^{-1}}$) projected by MAR forced by ACCESS1.3 (purple), CESM2 (green), CNRM-CM6-1 (orange) and NorESM1-M (blue) for all the Antarctic ice shelves (a), the ice shelves of the East Antarctic Sector (b), the West Antarctic Sector (c), and the Antarctic Peninsula (d) compared to mean summer ESM near-surface temperature (°C) over the 90°S-60°S.

The differences in projected melt and SEB in 2100 are partly linked to the ESM warming sensitivity. The latter is commonly expressed by the equilibrium climate sensitivity (ECS, see supplement in Zelinka et al. (2020) for CMIP5 and CMIP6 models). As suggested by their ECS, MAR forced by NorESM1-M (ECS of 2.8) and ACCESS1.3 (ECS of 3.55) project a lower future surface melt than the two other experiments. Nonetheless, ECS does not wholly explain the differences between the CESM2
(ECS of 5.15) and CNRM-CM6-1 (ECS of 4.9) experiments as the latter projects a larger surface melt increase. This could be explained by the greater regional warming over the Antarctic region simulated by CNRM-CM6-1 (+8.5°C vs 7.7°C for CESM2 in 2100 compared to 1981-2010). However, MAR forced by CNRM-CM6-1 still simulates a larger melt increase for the same warming rate than the other experiments (Fig. 2a). This highlights that other local physical mechanisms have to be involved in addition to ESM warming rates to explain the spread in future surface melt. Figure 2 further reveals that the WAIS exhibits the
highest spread in surface melt for a given warming rate, confirming that the main uncertainties in future Antarctic surface melt result from this region. We will therefore analyse the factors behind LWN, and more precisely behind LWD differences over the WAIS, focusing especially on the CNRM-CM6-1 and CESM2 experiments while keeping in mind their (relatively) close ECS and regional Antarctic warmings.

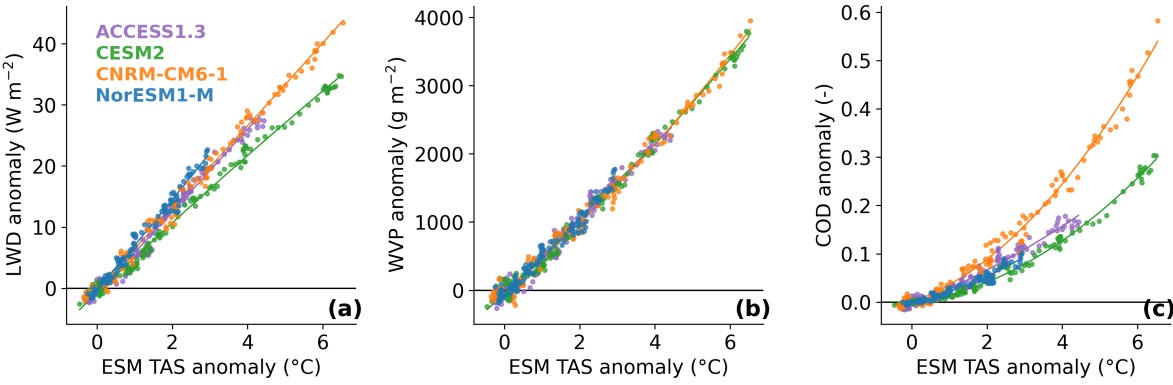

**Figure 3.** Changes in downwelling longwave fluxes ($W\,m^{-2}$) (a), Water Vapour Path ($g\,m^{-2}$) (b), and Cloud Optical Depth (-) (c) as projected by MAR forced by ACCES1.3 (purple), CESM2 (green), CNRM-CM6-1 (orange) and NorESM1-M (blue) over the ice shelves of the West Antarctic sector compared to mean summer ESM near-surface temperature (°C) over the 90°S-60°S.

### 3.2 Factors behind the differences in LWD over the West Antarctic ice shelves

The projected LWD increases in each experiment are mainly due to higher atmospheric temperature, larger greenhouse gas concentrations including water vapour, and optically thicker clouds. We perform our MAR projections using RCP8.5 for CMIP5 forcings and SSP5-8.5 for CMIP6 forcings. Despite differences in specific anthropogenic greenhouse gas concentrations, these two scenarios result in the same radiative forcing in 2100 (+8.5 $W\,m^{-2}$). We will therefore analyse the contribution of the remaining factors - atmospheric temperature, water vapour and cloud properties.

#### 3.2.1 Changes in atmospheric temperature and water vapour

For a similar warming rate, the differences in projected atmospheric temperatures and water vapour content only account for small differences in LWD. The increase in temperature of the atmosphere related to the sensitivity of the ESM forcing determines the absolute increases and differences in LWD (Fig. 3a). This is notably highlighted by the differences between MAR forced by NorESM1-M and the other experiments. However, temperature alone is not sufficient to explain the large LWD differences for the same warming rate (Fig. 3a). Approximating the atmosphere as a longwave-opaque and black body (see Sect. S4), we estimated the maximal potential contribution of the atmospheric temperature in summer over the present (1981–2010) and the end of the 21st century (2071–2100) in Table S1. For instance, we found that the future atmospheric temperature in MAR forced by CESM2 and CNRM-CM6-1 could not explain more than 31% of modelled future LWD differences (2.2 $W\,m^{-2}$ over to 7.1 $W\,m^{-2}$) over the ice shelves of the WAIS sector. Higher atmospheric water vapour content favour higher LWD but all MAR experiments project similar increases in water vapour for the same warming rate following the Clausius-Clapeyron relation (Fig.3b).

The absolute increases and differences in LWD are linked with the temperature of the atmosphere. The warming sensitivity of each ESM (as indicated by their ECS) influences the atmospheric temperature and water vapour content for a given future time period, explaining melt changes that are projected to be weak (NorESM1-M), intermediate (ACCESS1.3) or strong (CNRM-CM6-1 and CESM2) by 2100. Accordingly, the predominant factor contributing to melt differences is the warming projected by each ESM, highlighting the importance of multi-model projections for a better assessment of uncertainties. However, comparing our results for the same rate of warming (see above the respective ECS of CNRM-CM6-1 and CESM2 or their Antarctic warming) suggests that other physical processes are at play, such as cloud feedbacks, for explaining the large potential melt differences projected for the same rate of warming.

### 3.2.2 Changes in cloud properties

The contribution of clouds to LWD mainly depends on their own longwave emissivity. The latter can be modified by the COD, strongly affected by cloud phase. Furthermore, a larger cloud cover (CC) also favours larger LWD values even for unchanged physical properties such as cloud opacity and thickness. As an illustration the MAR experiments project a larger cloud cover over the Ross and Ronne-Filchner ice shelves and also more opaque clouds and and consequently a decrease in SWD.

The mean summer CC and COD are projected to increase over the WAIS during the 21st century (Fig 4). While MAR driven by ACCESS1.3, NorESM1-M, and CESM2 have similar CC increases (between ∼3% and ∼4%), the CNRM-CM6-1 experiment (i.e., with the strongest surface melt) reveals the largest cloud cover increase with 9% more frequent clouds during the austral summer. This is more than a factor of two compared to the other projections. Similarly, COD increases with a factor of ∼ 5 between the smallest (NorESM1-M) and the largest (CNRM-CM6-1) changes (Fig 4). While higher temperatures lead to larger COD increases, Figure 3c demonstrates that the future changes are not only a direct consequence of atmospheric warming. For instance, MAR driven by CNRM-CM6-1 simulates stronger changes in COD than other experiments for equivalent near-surface warming rates over the ice shelves. This again highlights the amplifying role of clouds as the main driver of surface melt for a given warming rate.

The relations expressed in Fig. 5 suggest that the sensitivity of the LWD increase would progressively stop for (very) large increases in COD. As these values are not reached before 2100 in our simulations, the future LWD increase is supposed to remain sensitive to cloud optical properties during the whole 21st century, including for high warming rates as projected by CNRM-CM6-1 and CESM2.

### 3.2.3 Changes in cloud particle phase and mass

MAR projects an increase in cloud particle contents and changes in phase distributions over the ice shelves that differ between the simulations, resulting in different cloud optical properties (Figs. 6, 7). While all the experiments start with similar IWP values (defined as the total ice and snow content in the whole atmospheric column), the increase is of different magnitude in each experiment with an almost fourfold increase between the lowest (7.3 $\mathrm{g\,m^{-2}}$ in NorESM1-M) and the highest (26.8 $\mathrm{g\,m^{-2}}$ in CESM2) changes. Similarly all experiments simulate an increase in LWP over the West Antarctic ice shelves in the

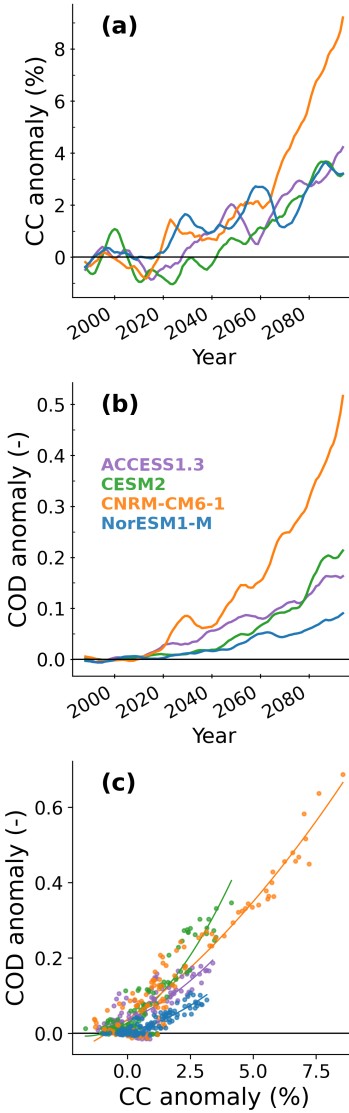

**Figure 4.** Changes in mean summer Cloud Cover (%) (a), mean summer Cloud Optical Depth (-) (b), changes in mean summer Cloud Optical Depth (-) as a function of changes in mean summer Cloud Cover (%) (c) projected by MAR forced by ACCESS1.3 (purple), CESM2 (green), CNRM-CM6-1 (orange), and NorESM1-M (blue) compared to the present summer climate (1981–2010) over the ice shelves of the west Antarctic sector. Values are averaged using a 10-year rolling mean.

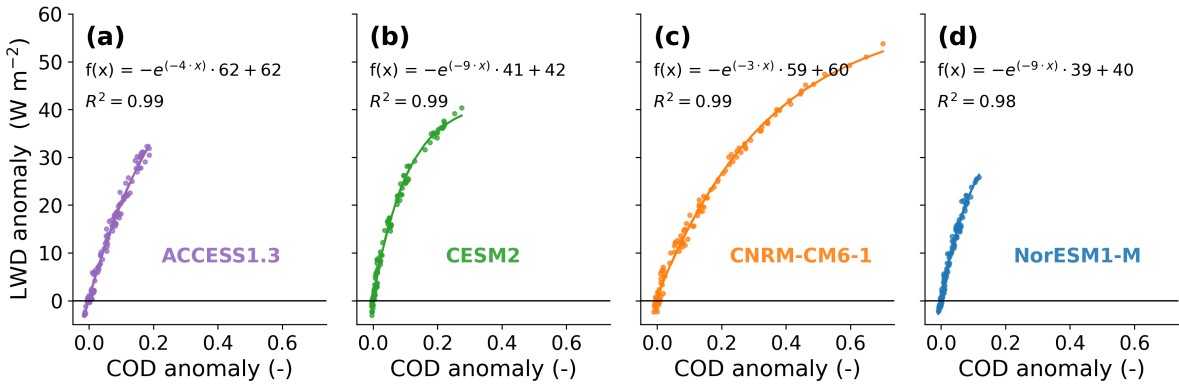

**Figure 5.** Summer LWD ($\mathrm{W\,m^{-2}}$) versus COD changes during summer (-) projected by MAR driven by ACCESS1.3 (a), CESM2 (b), CNRM-CM6-1 (c), and NorESM1-M (d) compared to the summer reference period (1981–2010). The exponential regression as well as corresponding determination coefficient ($R^2$, $p \ll 0.01$) is indicated for each experiment. A 10-year running mean has been applied.

future, but large differences persist between the changes. MAR driven by CNRM-CM6-1 projects a stronger increase in LWP (8 $\mathrm{g\,m^{-2}}$), 8 times larger than the increase in the NorESM1-M experiment (1 $\mathrm{g\,m^{-2}}$) over 2071–2100.

The different increases in LWP control the spread in projected LWD for a same warming rate. This results from the strong dependence of cloud emissivity on liquid water content (Stephens, 1984; Bennartz et al., 2013). While the CESM2 experiment suggests slightly larger changes in IWP than the CNRM-CM6-1 experiment, the latter projects more liquid-containing clouds (higher LWP) resulting in more opaque clouds (higher COD and then higher LWD) for the same warming rate. The CNRM-CM6-1 experiment tends to project larger increases in LWP over all the ice shelves than the other experiments for similar warming rates. However, the difference compared to the other experiments is only as large as over the WAIS as revealed by Fig. 7. This analysis highlights the strong influence of the cloud water phase for explaining melt differences projected for the same warming rate over the WAIS, a region we previously identified to control the future melt uncertainties.

The projected cloud phase differences are explained by the preferential increase of either water and rain droplets or ice and snow particles at a same warming rate. Over 2071–2100, both the vertically-averaged atmospheric changes in humidity and temperature projected by MAR driven by CESM2 and CNRM-CM6-1 are similar over the ice shelves of WAIS (Tab. S2). This enables a direct comparison removing the influence of global warming on potential differences. At the lateral boundaries, the CESM2 experiment reveals a stronger increase in specific humidity above 2000 masl than MAR forced by CNRM-CM6-1 (Fig. 8a). The pattern is opposite below 2000 masl, where the future CNRM-CM6-1 atmosphere is characterised by stronger low-level humidity advection. Supplementary maps (Fig. S5 and Fig. S6) illustrate that these discrepancies are relatively spatially uniform along the boundaries of MAR. However, our results suggest stronger differences over the WAIS and both the Ross and Filchner-Ronne ice shelves. High- and mid-level humidity advection favours the formation of snow particles in the CESM2 experiment (Fig. 8b), while low-level humidity advection, where the temperature is higher, leads to the formation of more water droplets in the CNRM-CM6-1 experiment (Fig. 8c). Favouring the formation of either snow (and ice) particles or

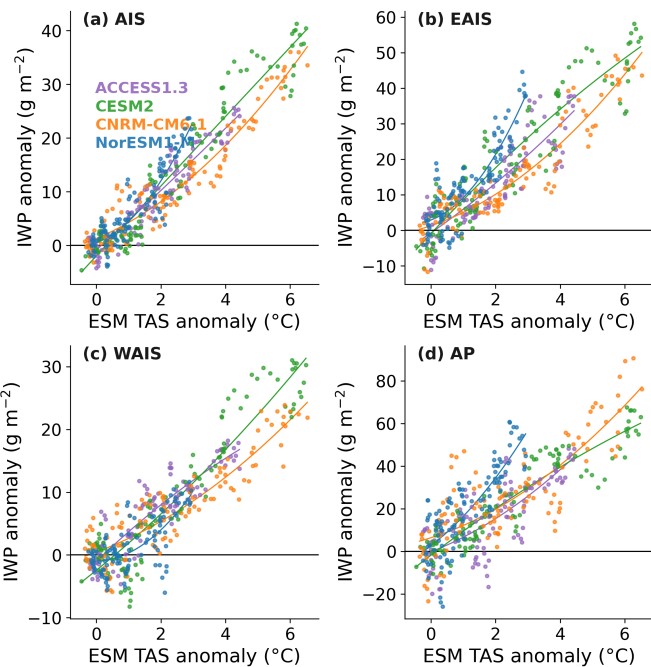

**Figure 6.** Changes in mean summer IWP $(\mathrm{g\,m^{-2}})$ as projected by MAR forced by ACCESS1.3 (purple), CESM2 (green), CNRM-CM6-1 (orange) and NorESM1-M (blue) for all the Antarctic ice shelves (a), the ice shelves of the East Antarctic Sector (b), the West Antarctic Sector (c), and the Antarctic Peninsula (d) compared to mean summer ESM near-surface temperature (°C) over the region 90°S-60°S.

water droplets when saturation is reached results in differences in IWP and LWP that further induces changes in LWD over the WAIS sector. The preferential future increase in low-level water droplets in the CNRM-CM6-1 experiment finally induces a stronger surface melt over the ice shelves than the CESM2 experiment despite a similar regional warming rate. Furthermore, the preferential increase in either cloud water droplets or snow particles also explains why MAR driven by CNRM-CM6-1 simulates more liquid precipitation than when driven by CESM2 and conversely for solid precipitation (see the Fig. 7 in Kittel

et al. (2021)).

### 3.3    Enhanced shortwave absorption and influence on surface albedo

The ground surface is projected to absorb more shortwave despite decreased SWD over all ice shelves. The SWD changes are determined by changes in cloud cover and properties. The MAR experiments project more opaque clouds and an increase in CC everywhere on the ice shelves. The noticeable exception is the AP, where CC is projected to decrease especially in the

CESM2 experiment. However, the COD effect dominates over the CC changes still leading to a decrease in SWD even on the AP. The excess energy at the surface warms and melts snow. This in turn promotes snow grain metamorphism that combined with refreezing of liquid meltwater, lowers the albedo and ultimately favours SWD absorption. This effect dominates over the decrease in SWD caused by the more numerous and also more opaque clouds, leading to an increase in SWN.

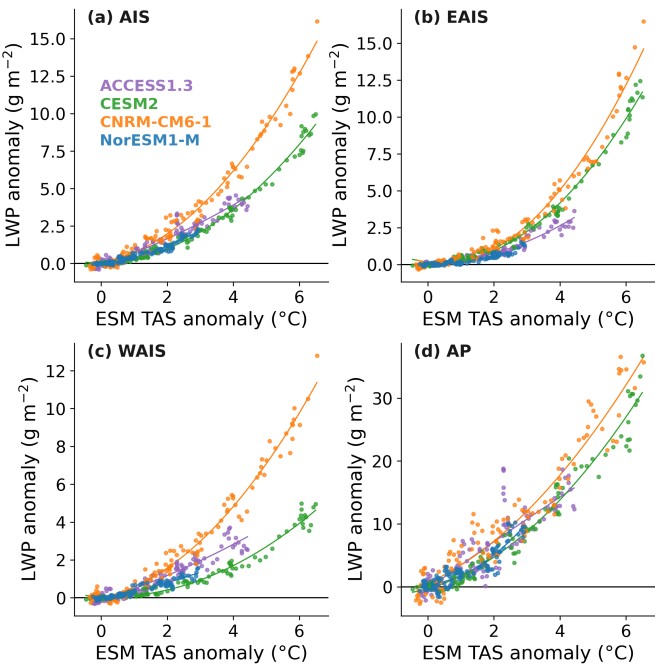

**Figure 7.** Changes in mean summer liquid water path $(\text{g m}^{-2})$ as projected by MAR forced by ACCES1.3 (purple), CESM2 (green), CNRM-CM6-1 (orange) and NorESM1-M (blue) for all the Antarctic ice shelves (a), the ice shelves of the East Antarctic Sector (b), the West Antarctic Sector (c), and the Antarctic Peninsula (d) compared to mean summer ESM near-surface temperature (°C) over the 90°S-60°S.

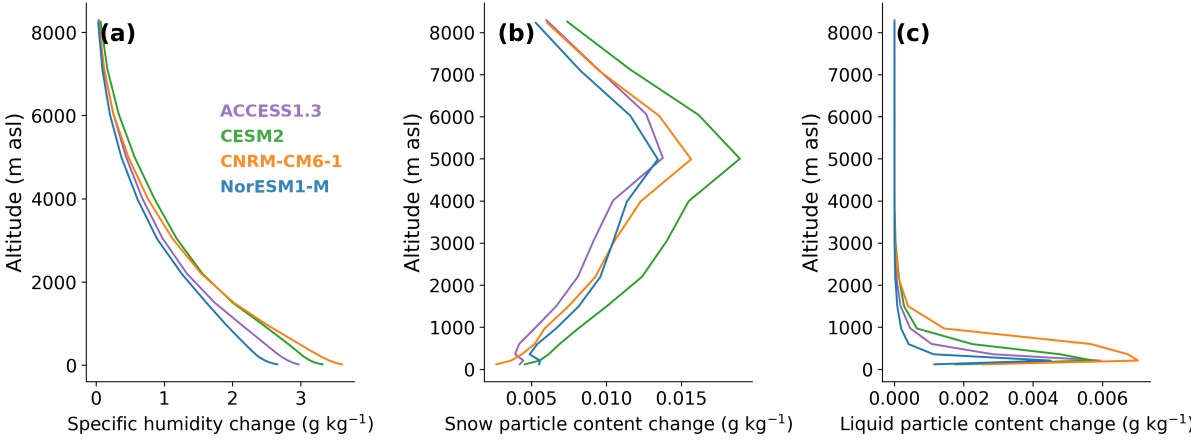

**Figure 8.** Changes in mean summer vertical specific humidity profiles over the boundaries (a), snow particle content (b), and water droplet particle content (c) $(\text{g kg}^{-1})$ over the ice shelves in 2071–2100 compared to 1981–2010 projected by ACCESS1.3 (purple), CESM2 (green), CNRM-CM6-1 (orange), and NorESM1-M (blue).

We compared the albedo decrease in MAR simulations to the forcing temperatures in the ESM. Figure 9 reveals that MAR forced by CNRM-CM6-1 projects a stronger albedo decrease over the WAIS sector associated with large warming rates compared to MAR forced by CESM2. This results from the discrepancies in cloud properties discussed above, leading to different melt rates and associated changes in albedo. While more liquid precipitation in some MAR experiments could contribute to further decrease the surface albedo, a sensitivity experiment in MAR forced by CNRM-CM6-1 where rainfall amounts were set to 0 reveals no difference with the original MAR CNRM-CM6-1 experiment. This is explained by the larger increase in melt compared to rain and then the predominant effect of the melt increase on the albedo decrease. This suggests that differences in liquid precipitation due to clouds do not further strengthen melt differences, at least for the precipitation rates projected by our different MAR experiments.

Finally, our projections also illustrate the competitive effects of clouds on solar radiation absorbed by the surface, as they reduce the surface albedo through enhanced LWD and melt but also reduce incoming energy by filtering SWD. Their influence on absorbed SWD mainly depends on the surface albedo but also on the rate at which SWD is projected to decrease due to an increase in CC and/or COD (Bintanja and van den Broeke, 1996). In warmer climates in which the albedo is projected to decrease, clouds could be more reflective than the ice-covered surface, as summer surface albedo is projected to decrease. These warmer conditions could reverse the summer cloud radiative effect, reducing melt, similarly as over the dark ablation zone of the Greenland Ice Sheet (Hofer et al., 2017; Wang et al., 2019), suggesting a growing importance of surface albedo in determining the future cloud radiative effect but also more generally SEB and melt changes over the AIS.

## 4  Discussion and conclusion

We investigate in this study the physical drivers of summer melt differences over the Antarctic ice shelves by 2100 between four dynamical downscaling of CMIP5 and CMIP6 ESMs with the polar-oriented regional atmospheric model MAR under the highest greenhouse gas concentration pathways (RCP8.5 and SSP5-8.5). Our results highlight the important role of clouds in amending future surface melt over the Antarctic ice shelves. The main differences in melt between our simulations arise from differences in LWN and SWN radiative fluxes. Among these fluxes, LWN is the most influential. Furthermore, we highlight the importance of cloud water content and phase to explain the differences in projected melt for a given warming. More liquid-water-containing clouds induce a stronger increase in LWD that enhances meltwater production but also favours SWD absorption due to the melt-albedo feedback, further increasing melt. Finally, we find that this preferential increase in water droplets results from a stronger increase in low-level humidity advection rather than high- and mid-level advection that tends to favour the formation of snow and ice particles.

While it is common to assess the Antarctic contribution to SLR associated with specific warming rates (e.g., Pattyn et al., 2018), liquid-containing clouds could lead to large uncertainties even for the same warming rate. For instance, the larger melt rate projected in the CNRM-CM6-1 experiment could lead to more areas susceptible to hydrofracturing compared to the CESM2 experiment despite a similar warming rate. In 2100, MAR driven by CNRM-CM6-1 projects that around 99% (76% over 2071–2100) of the Antarctic ice shelves could be vulnerable to surface melt-driven disintegration (Gilbert and Kittel,

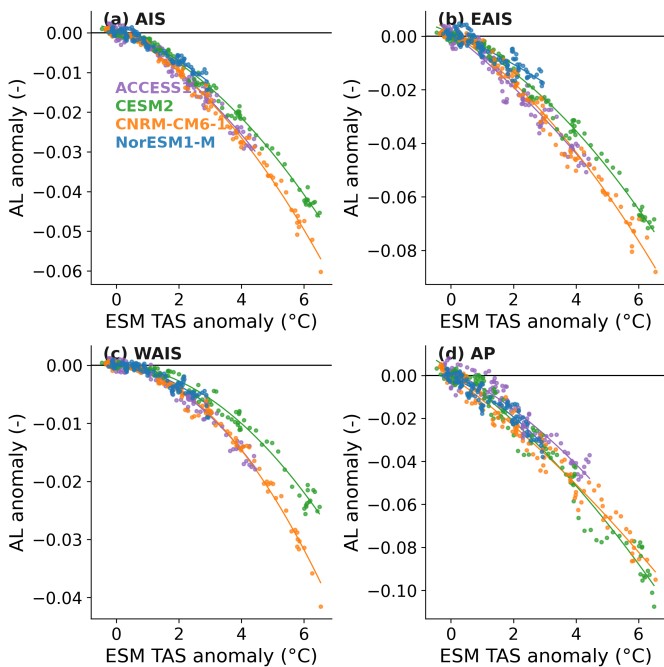

**Figure 9.** Changes in mean summer albedo (-) as projected by MAR forced by ACCES1.3 (purple), CESM2 (green), CNRM-CM6-1 (orange) and NorESM1-M (blue) for all the Antarctic ice shelves (a), the ice shelves of the East Antarctic Sector (b), the West Antarctic Sector (c), and the Antarctic Peninsula (d) compared to mean summer ESM near-surface temperature (°C) over the 90°S-60°S.

2021). Without the buttressing effect of these ice shelves, the Antarctic glaciers accelerate, increasing their discharge into the ocean and raising global sea level (Sun et al., 2016). This suggests that clouds are projected to have a strong effect on determining the Antarctic contribution to SLR.

While MAR projections reveal significant melt differences using different ESM forcings, we emphasise here that none of these projections is more plausible than any other and that the purpose of this study is, on the contrary, to highlight the physical factors that can lead to large uncertainties in Antarctic melt projections. The warming projected by the ESM forcing is the main factor controlling absolute melt differences, but we suggest that clouds and their phase as simulated in MAR are important factors contributing to the spread in melt and by extension surface mass balance projections of the AIS for the same warming

rate. Recent studies with MAR (Le Toumelin et al., 2021; Hofer et al., 2021) have revealed significant changes in LWD due to drifting snow, a process not modelled in our study, suggesting that drifting snow could further contribute to the spread in melt projections. Furthermore, MAR seems to underestimate the present summer LWP compared with CloudSat-Calipso estimates. Our study highlights the sensitivity of the future surface melt to liquid-containing clouds whose representation is considered a challenge for climate models in Antarctica (Listowski and Lachlan-Cope, 2017; Vignon et al., 2021). Future work should

improve the cloud representation (including in MAR) potentially leading to revised melt projections over the Antarctic ice shelves.

*Code and data availability.* The MAR code used in this study is tagged as v3.11.1 on https://gitlab.com/Mar-Group/MARv3.7 (MARTeam, 2021). Instructions to download the MAR code are provided on https://www.mar.cnrs.fr (MARmodel, 2021). The MAR version used for the present work is tagged as v3.11.1. The MAR outputs used in this study are available on Zenodo (https://doi.org/10.5281/zenodo.6406158; (Kittel, 2022). Other higher-frequency MAR results and Python scripts are also available upon request by email (ckittel@uliege.be).

*Author contributions.* CK designed the study, ran the simulations, made the plots, performed the analysis and wrote the manuscript. ChA, XF provided important guidance while all the authors (Ck, ChA, SH, CéA, NCJ, EG, LLT, ÉV, HG and XF) discussed and revised the manuscript.

*Competing interests.* The authors declare that they have no conflict of interest.

*Acknowledgements.* We acknowledge the World Climate Research Programme's Working Group on Coupled Modelling, which is responsible for CMIP, and we thank the climate modelling groups for producing their model output and making it available.

Computational resources have been provided by the Consortium des Équipements de Calcul Intensif (CÉCI), funded by the Fonds de la Recherche Scientifique de Belgique (F.R.S. – FNRS) under grant no. 2.5020.11 and the Tier-1 supercomputer (Zenobe) of the Fédération Wallonie Bruxelles infrastructure funded by the Walloon Region under grant agreement no. 1117545. This research has been supported by F.R.S.-FNRS, the Fonds Wetenschappelijk Onderzoek-Vlaanderen (FWO) under the EOS project no. O0100718F and under grant no. T.0002.16. C.Kittel and N.C. Jourdain have received funding from the European Union's Horizon 2020 research and innovation programme under grant agreement No 101003826 via project CRiceS (Climate Relevant interactions and feedbacks: the key role of sea ice and Snow in the polar and global climate system).

Finally, we would like to thank the Rajashree Datta and the other anonymous reviewers for their constructive remarks that helped to improve the paper.

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
