# Peer review of "Clouds drive differences in future surface melt over the Antarctic ice shelves"

_The Cryosphere, 2021_

## Author Comment (AC1)

**Reviewer #1**

We would like to thank the reviewer #1 for the comments and suggestions for producing new analyses that helped us to improve our manuscript. Please find our responses in blue below your comments. Italic text represents unchanged text from our manuscript while bold text highlights our suggested changes.

This manuscript is a well written paper that presents the potential uncertainties we might expect in future surface melt predictions due to the role of clouds. The manuscript is well constructed and uses state-of-the-art regional climate model data but simply passes many of the complexities related to the surface melt – albedo feedback. Before this study can be published the authors should consider a much more careful analysis of the contemporary surface melt production, including a robust evaluation of clouds and their phase differences. In addition, other influencers of the surface albedo feedback should at least be mentioned, such as the role of precipitation (snow + rain!) and the role of cloudy and clear-sky conditions. Below I will divide my comments into major and minor comments and hope to explain my concerns

**Major comments**

1. Surface melt and cloud evaluation

The major thing missing in this manuscript is a careful evaluation. Yes, the authors cite several of the studies evaluating MAR energy fluxes, SMB and surface melt rates. However, (future) surface melting is extremely sensitive, especially the role of clouds and their phases. See for instance King et al., 2015 and the study of one of the authors Gilbert et al., 2020. I would like to see more convincing evidence that contemporary melting is accurately modelled and for the right reasons (you yourself note the effects of compensating errors) and that the surface albedo feedback is (reasonably) correct. How does MAR represent surface melting in cloudy conditions and in clear-sky conditions? There are ample ways to improve and extend the surface melt evaluation with this paper. An example is Jakobs et al., 2020, which provides a state-of-the-art evaluation dataset publicly accessible against which the contemporary melt climate modelled by MAR can be tested.

Although we understand the concerns of the reviewer about the model ability to represent surface melting in different conditions, we think that a detailed evaluation as done in King et al. (2015) or Gilbert et al. (2020) is beyond the scope of our manuscript. Despite a real interest to evaluate MAR,  this requires a full paper dedicated to this topic (as is done in the papers the reviewer highlights, King et al. (2015) and Gilbert et al. (2020)) and a short study period (i.e one month), while the aim of our study is to understand what could be (in MAR) the physical drivers leading to large differences in melt projections by the end of the century. Furthermore, although partly using AWS observation, Jakobs et al. (2020) dataset remains a model product with its own biases and therefore cannot be used so directly as a reference for model evaluation. It would then require a more detailed comparison taking into account instrument uncertainties (up to 10% following Jakobs et al. (2020)) which can be locally higher than the daily biases of MAR fluxes (see Kittel (2021) and Hofer et al., (2021)). We acknowledge that a daily melt evaluation of MAR has not yet been accomplished, but please

note the yearly evaluation as presented in Kittel et al. (2021) is similar to the comparison of RACMO or QuikSCAT with the estimations provided in Jakobs et al. (2020).

However, we provide a "raw" daily comparison as suggested by the reviewer with Jakobs et al. (2020)' estimates (Fig. R1). It shows for all AWS a satisfactory representation of melt, with a reproduction of realistic patterns in melt time series. Furthermore, we note some differences between MAR and Jacobs et al. (2020) products, such as the absence of some relatively small melt events at AWS6 and AWS16 in MAR. We also focussed on one summer at AWS17 (between Larsen B and C) which did not reveal, at a first glance, a particular melt bias related to cloud cover (Fig. R2).

[Figure]

Fig. R1: Daily comparison of melt estimates from Jakobs et al. (2020) (blue) and simulated by MAR (orange) at several AWS locations.

In the same way, we compared AWS and MAR albedo (Fig. R3) showing a good temporal evolution of the modelled albedo during melt (and likely snowfall) events. This then suggests that the melt-albedo feedback is properly, at least qualitatively, captured by MAR over the present climate. Nonetheless, the comparison reveals that MAR tends to underestimate the surface albedo and that further investigation will be necessary to determine if it results from the underestimation of the fresh snow albedo, or biases in the snowfall amount (and snow thickness over ice at the surface), cloud cover, liquid water production or refreezing, or a combination of all these elements that influence MAR albedo. In the same way, the comparison should also take into account uncertainties in the measurements.

[Figure]

Fig. R2: Daily comparison of melt estimates from Jakobs et al. (2020) (blue) and simulated by MAR (orange) at AWS17 during the 2015–2016 summer with the cloud cover from Jakobs et al. (2020).

[Figure]

Fig. R3: Daily comparison of surface albedo from Jakobs et al. (2020) (blue) and simulated by MAR (orange) at AWS17 during the 2015–2016 summer with the cloud cover from Jakobs et al. (2020).

In addition, before any weight can be given to the modelled cloud fractions, contemporary cloud fractions should be evaluated in more detail. Examples to do this are the Van Tricht et al., 2016 and Lenaerts et al. 2017 papers (I'm not aware if updates of these products are available yet). They provide extensive Antarctica-wide products of liquid and solid water clouds, against which cloud fractions of MAR can be tested.

We provide a comparison of the mean summer IWP (Fig. R4) and LWP (Fig. R5) from Cloudsat-Calipso and MAR over 2010–2016 as also requested by #Reviewer2. Due to the

difference in grid resolution (2° for the CloudSat-Calipso product vs 35km for MAR), we compared each product on its own grid as interpolating one product on the other grid would increase the uncertainties.

[Figure]

Fig. R4: Mean summer Ice Water Path over 2010–2016 as deduced from Cloudsat-Calipso (Van Trich et al., 2016 and Lenaerts et al., 2017) (left) and as simulated by MAR forced by ERA5 (right).

[Figure]

Fig. R5: Mean summer Liquid Water Path over 2010–2016 as deduced from Cloudsat-Calipso (Van Trich et al., 2016 and Lenaerts et al., 2017) (left) and as simulated by MAR forced by ERA5 (right).

As revealed by Fig. R4, MAR tends to overestimate the IWP compared to the CloudSat-Calipso product while still presenting a similar spatial pattern. For instance, high values suggested by the CloudSat-Calipso product near the Peninsula, the Enderby Land

(and Cosmonaut Sea), and Wilkes Lands (and Mawson Sea) correspond to places where MAR simulates its highest IWP values. In the same way, low values of IWP can be found over the Ross and Weddell Seas in both products. MAR underestimates the LWP compared to the CloudSat-Calipso product, especially over the ocean although both products are more in agreement over the ice shelves. It is known that MAR underestimates LWP (Mattingly et al., 2020, Sedlar et al., 2020) and could be partly explained by a too strong vertical mixing (Inoue et al., 2021). Other studies (eg., Vignon et al. 2021) have also linked a poor representation of mixed-phase clouds over the Southern Ocean due to a poor representation of the turbulence-microphysics coupling as well as to an overall overestimation of the predicted concentration in ice nuclei particles (and ice crystal growth rate) in standard cloud parameterizations.

Despite having a correct agreement with ground observations (Lenaerts et al., 2017), the CloudSat-Calipso product is marred by several uncertainties related to the satellites such as revisit time, aboard measure instruments (Listowski et al., 2019), but also owing the assumptions made to retrieve ice and liquid water contents from remotely-sensing signals (such as the crystal mass-diameter relationship used to retrieve the mass concentration of cloud ice (Delanöe et al., 2014)). For instance, the satellite overpass (one every 2 days at 60°S following Listowski et al., 2019) can miss short events with high values of particle contents such as atmospheric rivers that are responsible for most snowfall amounts (Wille et al., 2021). This results in an underestimation of IWP. Furthermore, the lidar signal is strongly attenuated when meeting a liquid layer or thick ice cloud preventing a proper characterization of the cloud layers below. Similarly, the signal is affected by the ground clutter; no reliable measurement is possible below 1300m a.g.l. (Palerme et al., 2014) i.e. in an altitude range where most boundary-layer clouds occur We have also compared MAR cloud cover with CloudSat and ERA5 in Hofer et al., 2021. The comparison suggests a good agreement with ERA5, while an overestimation against CloudSat. This could come from MAR, the forcing (ERA5) or also the clutter effect and the underestimation of Cloudsat due to particles below 1300 m. The comparison thus suggests cloud biases but it is not very conclusive since one should include the observational uncertainties associated with the mass/size relation, the horizontal resolution, and the limited temporal (revisit time) and spatial (above 1300 m agl) coverage. One could for instance compute an IWP between 1300 m agl and the top of the atmosphere in MAR (and the satellite product), perform the comparison only at satellite overpass times (similarly to the evaluation of Palerme et al. 2014 for the Cloudsat precipitation and ERA-Interim), and complement the evaluation with ground-based observation comparisons. We however think that this more correct and comprehensive comparison is beyond the scope of this paper, and we stress that this first comparison with the Cloudsat-Calipso product should be taken with caution.

All this analysis does not mean that future MAR liquid clouds are unrealistic. If there is a bias in the current model estimates (likely but should be refined with a proper comparison knowing the uncertainties of the observational reference used), it affects all simulations in an equivalent way, and its influence is likely damped when looking at differences betweens different downscallings of ESM, all produced with the same MAR physics. We therefore think it should not have an influence on the main results of this paper: MAR liquid clouds generate differences in melt projected over the ice shelves despite a same warming rate. As mentioned L300 in our manuscript, we already emphasise that none of the MAR projections

is more plausible than any other and that the purpose of this study is to understand which physical factors lead to large uncertainties in Antarctic melt estimates projected by the RCM. We clarify and emphasise this aim in the introduction modifying L31-48:

*Surface melting in Antarctica is currently predominantly limited to Antarctic ice shelves \citep{trusel2013,van2018, Agosta2019}, the floating extensions of the grounded ice sheet. Surface melt can damage the ice shelves \citep{Lhermitte2020}, potentially initiate their collapse \citep{van2005} and increase the Antarctic contribution to sea level rise (SLR) through a speed-up in glacier flow \citep{Scambos2014}. Little is known about how cloud-related uncertainties will influence the future climate and surface mass balance projections over the Antarctic ice shelves.*

*Quantifying the influence of clouds on the SEB remains challenging over bright surfaces in high latitudes. This is particularly true over the AIS where observations are scarce and expensive to maintain \citep{bromwich2012, boucher2013}. From a modelling perspective, stronger positive cloud feedbacks over the southern ocean result in higher equilibrium climate sensitivities in Earth System Models (ESMs) from the recent 6th phase of the Coupled Model Intercomparison Project (CMIP6) than in the earlier 5th phase \citep{Zelinka2020,Wyser2020,wang2021}. Furthermore, ESMs usually lack the necessary spatial resolution and underlying physics to resolve the small floating ice shelves. For instance, coarse-resolution ESMs tend to project a lower decrease in SMB, and then likely a lower increase in melt compared to high-resolution regional projections \citep{kittel2021}. This highlights the need for a more detailed quantification of the future cloud effects with high-resolution and polar-oriented models to evaluate uncertainties related to cloud properties on the projected Antarctic surface melt and resulting SLR contribution.*

*To understand how the SEB drives the differences in future summer surface melt over the Antarctic ice shelves, we force the regional climate model "Modèle Atmosphérique Régional" \citep[MAR,][]{Gallee1994} with four ESMs from the CMIP5 (ACCESS1.3 and NorESM1-M) and CMIP6 (CNRM-CM6-1, CESM2) database using the highest greenhouse gas concentration pathways (respectively RCP8.5 and SSP585).*

To

**However, quantifying the influence of clouds on the SEB remains challenging over high latitudes. This is particularly true over the AIS where observations are scarce and expensive to maintain \citep{bromwich2012, boucher2013}. From a modelling perspective, stronger positive cloud feedbacks over the southern ocean result in higher equilibrium climate sensitivities in Earth System Models (ESMs) from the recent 6th phase of the Coupled Model Intercomparison Project (CMIP6) than in the earlier 5th phase \citep{Zelinka2020,Wyser2020,wang2021}. This might explain why CMIP6-based projections suggest more changes over the Antarctic Ice Sheet, and especially a higher increase in melt over the margins \citep{Kittel2021}.**

**Little is then known about how cloud-related uncertainties and in general SEB will influence the future climate and surface mass balance projections over the Antarctic ice shelves. Surface melting in Antarctica is currently predominantly limited to Antarctic ice shelves \citep{trusel2013,van2018, Agosta2019}, the floating extensions**

*of the grounded ice sheet. Surface melt can damage the ice shelves, potentially initiate their collapse \citep{van2005} and increase the Antarctic contribution to sea level rise (SLR) through a speed-up in glacier flow \citep{Scambos2014}. While melt can be derived from temperatures \citep{Trusel2015}, large differences in melt changes can be projected even for a same warming rate \{Kittel2021} that could lead to significant uncertainties in hydrofracturing risk \citep{gilbert2021}.*

*With the aim of understanding what could be the physical drivers of the SEB leading to large differences in melt projections over the Antarctic ice shelves, we force the regional climate model (RCM) "Modèle Atmosphérique Régional" \citep[MAR,][]{Gallee1994} with four ESMs from the CMIP5 (ACCESS1.3 and NorESM1-M) and CMIP6 (CNRM-CM6-1, CESM2) database using the highest greenhouse gas concentration pathways (respectively RCP8.5 and SSP585).*

We will also mention the biases in the representation of clouds as simulated by MAR (L54-66) (while removing the comparison of the biases with the radiative forcing due to GHC as the comparison is not appropriate since it compares values at the tropopause for the whole Earth against values at the surface for a single continent).

*The model has been thoroughly evaluated over the AIS against near-surface observations from automatic weather stations (AWSs) \citep{datta2018, Mottram2020, kittel2021, Amory2021} including radiative fluxes \citep{Letoumelin2020, kittelthesis}, SMB measurements \citep{Kittel2018, Agosta2019, Donat2020interannual, Mottram2020, kittel2021}, melt estimates derived from both satellites \citep{datta2018,Donat2020interannual} and AWSs \citep{kittel2021}. MAR underestimates summer SWD by -6.9~\unit{W}\,\unit{m^{-2}} and LWD throughout the year by -9.9~\unit{W}\,\unit{m^{-2}} \citep{kittelthesis}. While these biases seem significant compared to the future radiative forcing increase due to greenhouse gas concentration in 2100 \citep[+8.5~\unit{W}\,\unit{m^{-2}}in RCP8.5 and SSP585,][]{ONeill2016}, it is important to note that MAR correctly represents present Antarctic surface melt and near-surface temperatures \citep{kittel2021}. This suggests a correct representation of the SEB through compensating turbulent fluxes and in general compensating errors whose impacts on the future SEB and melt is difficult to assess. Furthermore, this study aims to explain the projected spread in melt illustrated in previous studies using MAR \citep{gilbert2021, kittel2021} rather than expanding on possible sources of misrepresentation of radiative fluxes in pre-existing simulations.*

To

*The model has been thoroughly evaluated over the AIS against near-surface observations from automatic weather stations (AWSs) \citep{datta2018, Mottram2020, kittel2021, Amory2021} including radiative fluxes \citep{Letoumelin2020, kittelthesis}, SMB measurements \citep{Kittel2018, Agosta2019, Donat2020interannual, Mottram2020, kittel2021}, melt estimates derived from both satellites \citep{datta2018,Donat2020interannual} and AWSs \citep{kittel2021}. MAR underestimates summer SWD by -6.9~\unit{W}\,\unit{m^{-2}} and LWD throughout the year by -9.9~\unit{W}\,\unit{m^{-2}} \citep{kittelthesis}.*

*It is important to note that MAR correctly compares with recent melt estimations and near-surface temperature observations \citep{kittel2021}. This suggests a satisfactory representation of the SEB likely due to compensating turbulent fluxes whose impacts on the future SEB and melt is difficult to assess.* **A first comparison (Fig.~S1) with CloudSat Calipso \citep{van2tricht2016, Lenaerts2017} suggests that MAR underestimates the liquid water path (LWP) but overestimates IWP (Fig.~S2) around Antarctica, which has also been reported by other studies over the Arctic \citep{Mattingly2020}. This underestimation should therefore be investigated in more depth in a separate study (with a better comparison) to understand this bias and improve the model. However, this bias affects all simulations in an equivalent way, and its influence is likely removed in comparisons between different downscallings of ESMs, all produced with the same MAR physics. This should not preclude an explanation of the physical drivers behind the projected spread in melt illustrated in previous studies using MAR \citep{gilbert2021, kittel2021} but should be kept in mind when discussing the plausibility of these projections.**

Finally, we suggest to adapt the last sentences of our manuscript (see our response to your major comments #5).

2. The role of snowfall and rain

This study solely looks at the radiative fluxes and their role in influencing surface melt production. However, especially in a future climate, precipitation rates are also going to increase, including a change in the fractions of rain and snow. Where obviously rain should be added to the surface melt rates (is it?), rain and snow also affect the melt-albedo feedback in several ways. This effect should be discussed. As of now, snow- or rainfall is not even mentioned once in the manuscript, while its relation to clouds is rather obvious. Additionally, a discussion can be added how future precipitation itself is influenced by cloud fraction, as it is likely just as important for ice shelf stability in a future climate as surface melting is.

The role of snowfall on albedo was not originally discussed as Shortwave Net contribution is relatively close for a same warming rate suggesting that the effect of the surface albedo decrease is compensated by the "parasol" effect of clouds (also stronger in the experiment where the surface albedo decrease the most, ie MAR forced by CNRM-CM6-1). The regional analysis requested by the reviewers sheds light on discrepancies in the SWN contributions so that we will discuss the albedo differences where it matters (See our response to #R1.3), as for instance over the Antarctic Peninsula.

Concerning rain, let us first clarify the different kinds of liquid water at the surface in the model: melting of snow at the surface, rain, or remaining water that cannot percolate or refreeze due to the snowpack (runoff in MAR) (or we did not understand what the reviewer meant by "rain should be added to the surface melt rates"). The amount of melt production results from the excess in energy in the SEB and is therefore an additional, different process to the rain leading to the presence of liquid water at the surface. Both melt and rain (i.e. liquid water) add up to determine the amount of water that percolates, (re)freezes, or is in excess and then considered to be runoff. The future effect of both melt and rain on runoff and then on SMB is discussed in Kittel et al. (2021). We here focus on the differences in melt

production only while as you mentioned, we also found a link between more melt due to liquid clouds, and more rain due to liquid clouds for a same warming (see L257-L262).

However, the differences in rain for the same warming between our experiments have likely no influence on melt differences, suggesting a low importance of the rain-albedo feedback (in MAR and for the warming range in our projections). This is explained by the stronger increase in melt amount compared to the increase in rain and the albedo parameterisation in the model. Once the snow grain metamorphism has lowered the albedo up to 0.7 (which often occurs during melt events, see Fig. R3), the "snow-firn" albedo is computed as a linear transition between 0.7 and 0.55 depending on the density.

The theoretical development in Donat-Magnin et al., 2021 (Appendix B1 and B2) highlights that given melt and rain amounts of the same magnitude, melt will more strongly favour the densification of the snow-firn (or the reduction in firn air content) than rain, as melt (not rain) consumes the firn. Since the increase in melt is between 7 and 10 times stronger than the increase in rain over 2071–2100 (Kittel et al., 2021), it explains why the rain-albedo effect cannot have a strong influence on the increase in melt and therefore on melt differences.

[Figure]

Fig. R5: Summer melt over the ice shelves projected by MAR forced by CNRM-CM6-1 (reference experiment in blue), and where we modify rain and snow before forcing the surface scheme. Orange line refers to the experiment with no rain, green and red respectively where 25% and 50% of snow are converted into rain.

Furthermore, we performed sensitivity experiments to more thoroughly address the reviewer's concern. We set rain to 0 (SS (RF=0)) or convert a fraction of snow into rain (25% hereafter referred to as SS(25%(SF) to RF) and 50% referred to as SS(50%(SF) to RF) before forcing the surface scheme. We performed these experiments over December 2094-2100 using CNRM-CM6-1. We did not find any differences compared to the reference run where we remove the rain in the surface scheme or when 25% of the snow is converted into rain (Fig R6). Only the experiment with the largest perturbations (where half of the snow becomes rain) suggests a significant increase in melt. In this specific experiment, the amount of rain is greater than the amount of snow, so that the rain vs snow ratio is larger than the ratio in Donat-Magnin et al. (2021, Appendix B1) leading to a strong decrease in

albedo while the albedo remains unchanged in the other experiments (Fig R6 and Fig R7). Although we haven't performed experiments where rain is converted into snow, we are confident that it will not lead to different conclusions as the increase in melt is much larger than the increase in rain and snow combined.

[Figure]

Fig. R6: Summer mean albedo over the ice shelves projected by MAR forced by CNRM-CM6-1 (reference experiment in blue), and where we modify rain and snow before forcing the surface scheme. Orange line refers to the experiment with no rain, green and red respectively where 25% and 50% of snow are converted into rain.

[Figure]

Fig. R7: Anomaly in summer mean albedo (%) between the experiment where 25% of snow is converted into rain and the reference run over 2094-2100.

The theoretical development and the sensitivity experiments suggest that differences in rain do not contribute to differences in melt. This does not mean that the rain-albedo feedback

cannot increase melt, but that the melt-albedo feedback is much stronger than the rain-albedo feedback up to 2100 or 8.5°C following the regional warming of CNRM-CM6-1. We then think there is no need to discuss the rain relationship with cloud fraction further and refer to Kittel et al., 2021 for precipitation changes but suggest to include this analysis in the new Section 3.3 that discuss (regionally) the importance of the albedo. (See the end of our response to #R1.3).

3. Regional climate

Throughout the manuscript ice sheet wide averages are used, even though about 30% to 50% of all (contemporary) surface melt occurs in the Antarctic Peninsula. Not only will the numbers presented therefore not be completely representative of all Antarctic ice shelves and the Antarctica Peninsula climate will dominate your results, you are also losing a lot of the local climate signals by averaging this way.

It is known (King et al., 2015 and other studies) that the surface energy budget (and hence melting) is heavily influenced by the partitioning of ice/water cloud particles, and that this process is very uncertain (like is concluded by yourselves) and very local. Some attention should therefore be spent on how these processes are governed elsewhere in Antarctica; e.g. do the same sensitivities exist over other ice shelves than Larsen C ice shelf? Some regional case studies should be performed to strengthen the conclusions in this study.

As also requested by Reviewer #2, we present the same analysis but per regions together with maps. We suggest changing most of the figures to include these more detailed results, and list all suggested changes in the manuscript associated with each figure. The message of the manuscript is unchanged (i.e., the influence of clouds and especially liquid-containing clouds to explain projected melt differences) but will take into account regional differences.

[Figure]

Fig. R8 : Cumulative surface melt (Gt) and SEB anomalies (Wm⁻²) over the Antarctic ice shelves. First row (a-e) shows the cumulated integrated surface melt and averaged SEB components over the whole Antarctic ice shelves, while second row (f,g,h,i,j) is for the Antarctic Peninsula, third row for East Antarctic sector including Ross and Ronne-Filchner ice shelves (k,l,m,n,o) and fourth row (p,q,r,s,t) for the West Antarctic sector. Second to firth columns represent the cumulated changes for each SEB component (green : net shortwave, orange : net longwave, purple : sensible heat, blue : latent heat) for each MAR simulations (second row : forced by ACCESS1.3, third row : CESM2, fourth row : CNRM-CM6-1, fifth row : NorESM1-M)

New section "3.1 Contributions to summer melt increase" will conserve its first paragraph that presents the values averaged for the whole Antarctic ice shelves while we suggest to replace L138-L165:

*Similarly, our MAR experiments project different melt increases over each region depending on the forcing ESM. Between the lowest and the highest increases, we found a factor of $\sim$2.5 over the Antarctic Peninsula (AP) (Fig.~\ref{ts_shelf}f), $\sim$4.4 over the West Antarctic Ice Shelves (EAIS) (Fig.~\ref{ts_shelf}k), and a factor of $\sim$5 over the East Antarctic Ice Shelves (WAIS) where we also included Ross and Ronne-Filchnner ice shelves (Fig.~\ref{ts_shelf}p). While the NorESM1-M and the ACCESS1.3 experiments project different increases over each region, the CNRM-CM6-1 and CESM2 experiments mostly only differ over the WAIS. There is indeed a factor $\sim$1.6 between these two projections despite a similar ESM warming. The WAIS (with Ross and Ronne-Filchnner) appears to be a region of major uncertainties as the differences in that specific sector dominates the Antarctic signal. Before discussing the SEB drivers leading to large differences of melt increase over the WAIS, we will firstly analyse the two other sectors because the changes in the energy balance (and then associated processes) are different in each region.*

*Over the AP, all flux changes are projected to positively contribute to the melt increases. MAR projects an individual similar positive contribution of radiative fluxes (longwave net (LWN) and shortwave net (SWN) for each experiment except when forced by CESM2 where the increase in SWN is stronger than in LWN. The relatively lower increase in LWN in this experiment results from the competitive effect of more opaque clouds (higher optical depth), but decreased cloud cover over the AP (Fig.~S3). These changes in cloud cover also contribute to decreasing snow precipitation \citep{Kittel2021}. The combination of increased melt and decreased snowfall lead to a large decrease in the albedo (Fig.~S4), explaining the higher contribution of SWN in the MAR forced by CESM2 simulation (Fig.~\ref{ts_shelf}p). It is interesting to note that the positive contribution of both turbulent fluxes is specific to the ice shelves of the Antarctic Peninsula. Recent studies \citep{Kuipers2012,vanWessem2014, Kuipers2018, Datta2019} have suggested that warm air advections (notably during foehn events) are an important source of energy over the Peninsula producing strong melt over the present climate. MAR simulations project a strong local warming due to warmer and moister air advections inducing higher precipitation \citep{kittel2021} but also larger melt rates. Since the snow/ice-covered surface cannot warm higher than the melting temperature, warmer air advections also increase the thermal inversion and then increase SHF. The positive SHF anomalies over the Peninsula even become dominant over the whole averaged Antarctic ice shelves (Fig.~\ref{ts_shelf}c-d) after 2060 when MAR is forced by CNRM-CM6 and CESM2.*

*The melt increase over the EAIS is projected to be dominated by the increase in radiative fluxes and essentially SWN. The MAR forced by NorESM1-M simulation excepted, all experiments project a stronger increase in SWN than LWN with a factor between $\sim$1.7 to $\sim$3.7. The large increase in SWN results from the albedo decrease (Fig.~S4) due to melt (and associated melt-albedo feedback) as snowfall are*

*projected to increase over the ice shelves of this sector \citep{Kittel2021}. The melt-albedo feedback also explains the relative high contribution of LWN compared to SWN in the NorESM1-M experiment as melt is likely too weak to actually trigger it taking into account the increase in fresh snow.*

*The EAIS sector including the two largest ice shelves drive the Antarctic-scale differences in projected melt. Following all MAR projections, the radiative fluxes explain the increase in melt while turbulent fluxes have a negative contribution. However, only LWN is projected to strongly increase and explains uncertainties in melt. The SWN contribution of MAR forced by CNRM-CM6-1 and CESM2 (and to a lesser extent ACCESS1.3) is almost equivalent, whereas the CNRM-CM6-1 experiment projects a much larger ($\sim$twice as large as) increase in LWN than all the other simulations. MAR projects an increase in cloud cover (Fig.~S3) enhancing LWN but is not sufficient to explain the projected differences (see hereafter). It is important to note that results in this sector are mostly driven by the Ross and Ronne-Filchnner ice shelves due to their superficies.*

*The contribution of a few specific events to produce melt over the Antarctic ice shelves is projected to change. We compared the amount of mean melt produced during the strongest summer events (daily melt above the p95 of the climate period) to the mean total summer amount of melt for the present period (1981-2010) and the future period (2071-2100). Over present-day conditions (Fig.~S6), this ratio is high (higher than 80\%, the peninsula excepted) suggesting that melt mainly occurs during specific events such as atmospheric rivers \citep{Wille2019}. On the contrary, all the MAR experiments project a much lower contribution of these specific events in the total summer melt (Fig.~S7). This suggests that melt will occur during more days in the whole summer.*

We then suggest changing L166-L177 (with update of the original Fig. ~S4 that will be moved into the main manuscript) by

*The differences in projected melt and SEB in 2100 are partly linked with the warming sensitivity of each forcing ESM. As suggested by the global response of an ESM to increase in greenhouse gas concentration or equilibrium climate sensitivity (ECS, see supplement in \citet{Zelinka2020} for CMIP5 and CMIP6 models), MAR forced by NorESM1-M (ECS of 2.8) and ACCESS1.3 (ECS of 3.55) project a lower future melt than the two other experiments. Nonetheless, ECS does not wholly explain the differences between the CESM2 (ECS of 5.15) and CNRM-CM6-1 (ECS of 4.9) experiments as the latter suggests a larger melt increase. This could be explained by accounting for the ECS capturing the greater warming over the Antarctic region simulated by CNRM-CM6-1 (+8.5°C vs 7.7°C for CESM2 in 2100 compared to 1981-2010). However, MAR forced by CNRM-CM6-1 still simulates a larger melt increase for the same warming rate than the other experiments (Fig.~\ref{meltvstas}a). This highlights that although model ECS contributes most strongly to uncertainty in melt and SEB, other local physical mechanisms have to be involved in addition to ESM warming rates. Figure~\ref{meltvstas} further confirms the highest spread for melt changes caused by ESM warming over ice shelves of the WAIS compared to other regions, making the WAIS the main region of uncertainty in our simulations. We will therefore analyse the factors behind the LWN, and more*

*precisely behind LWD differences over the WAIS, focusing especially on the CNRM-CM6-1 and CESM2 experiments having in mind their (relatively) close ECS and regional Antarctic warmings.*

[Figure]

Fig.R9: Mean summer melt changes compared to mean summer ESM near-surface temperature over the 90°S-60°S for all the Antarctic ice shelves (a), the ice shelves of the East sector (b), the West sector (c), and the Antarctic Peninsula (d) as projected by MAR forced by ACCES1.3 (purple), CESM2 (green), CNRM-CM6-1 (orange) and NorESM1-M (blue).

Section "3.2.1 Changes in atmospheric temperature and water vapour" (L185-198) will remain unchanged except we will update Table S4 based on values over WAIS. Note that the new version of FigS5 will be in the manuscript replacing the original FigS5, FigS6 and FigS7 and showing local comparisons with the forcing ESM temperature.

Section "3.2.2 Changes in cloud properties" will be also updated with the values of the WAIS sector.
L200-208 will become:

The cloud contribution to LWD mainly depends on their own longwave emissivity. The latter can be modified by the COD and therefore cloud phase. Furthermore, a larger cloud cover also favours larger LWD values even for unchanged physical properties (ie, COD). The MAR experiments project a larger cloud cover **over the Ross and Ronne-Filchnner ice shelves (Fig.~S3)** and also more opaque clouds (Fig.~\ref{cloudslwd}) and a decrease in SWD.

The mean summer cloud cover (CC) and COD are projected to increase during the 21st century (Fig~\ref{cloudslwd}). While MAR driven by ACCESS1.3, NorESM1-M, and CESM2 have similar CC increases (**between $\sim$3\% and $\sim$4\%)**, the CNRM-CM6-1 experiment (ie., with the strongest melt) reveals the largest cloud cover increase with **9\%** more frequent clouds during the austral summer. This is more than a factor of two compared to the other projections. In the same way, COD increases starting from $\sim$2020 with a factor$\sim5$ between the smallest (NorESM1-M) and the largest (CNRM-CM6-1) increases.

**The relations expressed in Fig.~3 suggest that the sensitivity of the LWD increase would progressively stop for (very) large increases in COD as the cloud emissivity approaches 1.** As these values are not reached before 2100 in our simulations, the future LWD increase is supposed to remain sensitive to cloud optical properties during the whole 21st century, including for high warming rates as projected by CNRM-CM6-1 and CESM2. While higher temperatures lead to larger *COD increases, Figure~S demonstrates that the future changes are not only a direct consequence of atmospheric warming. For instance, MAR driven by CNRM-CM6-1 simulates stronger changes in COD than other experiments for equivalent near-surface warming rates over the ice shelves. This again highlights the predominant influence of the ESM warming as the main driver of melt differences but also the amplifying role of clouds.*

Section 3.2.3 "Changes in cloud particle water phase and mass" will also take into account the regional analysis over the WAIS sector (with minor updates of the Table S2 and original figures therein).

L332-L360:
*MAR projects an increase in cloud particle contents and changes in phase distributions over the ice shelves that differ between the simulations, resulting in different cloud optical properties (Fig.~\ref{cloudicepath}, Fig.~\ref{cloudlwppath}). Over 2071--2100, the summer mean solid water path (SWP, the mean total amount of ice and snow content in the atmosphere averaged for every summer) increases similarly among experiments with anomalies between **7.3** \unit{g}\,\unit{m^{-2}} and **26.8** \unit{g}\,\unit{m^{-2}} which represents a factor of **3.7** between the lowest (NorESM1-M) and the highest increase (CESM2). This increase in the CESM2 experiment represents an increase of +37\% compared to present values and does not result from a specific underestimation of this experiment over the present climate, as all the experiments starts with similar SWP values around $\sim$~\unit{g}\,\unit{m^{-2}}. While all projections simulate a higher liquid water path (LWP, equivalent of SWP for water droplet content) in the future, large differences persist in the anomalies. MAR driven by CNRM-CM6-1 projects a stronger increase in LWP (**8.0** \unit{g}\,\unit{m^{-2}}) that is **8 times** larger than the increase in the NorESM1-M experiment (**1.0** \unit{g}\,\unit{m^{-2}}) over 2071--2100.*

*The different increases in LWP control the spread in projected LWD for a same warming rate. This results from the strong dependence of cloud emissivity on their liquid water content \citep{stephens1984, bennartz2013}. While the CESM2 experiment suggests slightly larger changes in SWP than the CNRM-CM6-1 experiment, the latter projects more liquid-containing clouds (higher LWP) resulting in more opaque clouds (higher COD and then higher LWD) for the same warming rate (Fig.~\ref{cloudicepath}, Fig.~\ref{cloudlwppath}).* **The CNRM-CM6-1 experiment tends to project larger increases in LWP over all the ice shelves than the other experiments for similar warming rates. However, the difference compared to the other experiments is only as large as over this specific region (ie, the WAIS).** *This analysis highlights the strong influence of the cloud water phase for explaining melt differences projected for the same warming rat***e over the WAIS, a region we previously identified to control the future melt uncertainties.***

*The projected cloud phase differences are explained by the preferential increase of either water and rain droplets or ice and snow particles at a same warming rate. Over 2071--2100, both the vertically-averaged atmospheric changes in humidity and temperature projected by MAR driven by CESM2 and CNRM-CM6-1 are similar over the ice shelves of WAIS (Tab.~S2). This enables a direct comparison removing the influence of global warming on potential differences.*

*At the lateral boundaries, the CESM2 experiment reveals a*stronger increase in specific humidity above 2000~\unit{m asl} than MAR forced by CNRM-CM6-1. The pattern is opposite below 2000~\unit{m asl}, where the future CNRM-CM6-1 atmosphere is characterised by stronger low-level humidity advection (Fig.~\ref{veticalqqp}a).* **Supplementary maps (Fig. S and Fig. S) illustrate that these discrepancies are relatively spatially uniform around the boundaries of MAR. However, our results suggest stronger differences over the WAIS and the large ice shelves (Ross and Filchnner-Ronne).** *High- and mid-level humidity advection favours the formation of snow particles (Fig.~\ref{veticalqqp}b), while low-level humidity advection, where the temperature is higher, leads to the formation of more water droplets (Fig.~\ref{veticalqqp}c). The formation of either snow (and ice) particles (CESM2) or water droplets (CNRM-CM6-1) when saturation is reached results in differences in SWP and LWP that further induces changes in LWD over the WAIS sector. The preferential future increase in low-level water droplets in the CNRM-CM6-1 experiment finally induces a stronger surface melt over the ice shelves than the CESM2 experiment despite a similar regional warming rate. Furthermore, the preferential increase in either cloud water droplets or snow particles also explains why MAR driven by CNRM-CM6-1 simulates more liquid precipitation than when driven by CESM2 and conversely for solid precipitation (see the Fig.~7 in \citet{kittel2021}).*

Finally, **we suggest to modify** **Section 3.3 Enhanced SWD absorption due to clouds (and rename it "Enhanced shortwave absorption and influence on the albedo")**

***The surface is projected to absorb more shortwave despite decreased SWD over all ice shelves. The SWD changes are determined by changes in cloud cover and properties. Over all the Antarctic ice shelves, the MAR experiments project more opaque clouds and an increase in CC almost everywhere. The noticeable exception is the AP, where CC is projected to decrease especially in the CESM2 experiment. However, the COD effect dominates CC changes still leading to a decrease in SWD***

*even on the AP. The excess energy at the surface warms and melts snow. This in turn promotes snow grain metamorphism that combined with refreezing of liquid meltwater, lowers the albedo and ultimately favours SWD absorption. This effect dominates over the decrease in SWD caused by the more numerous and also more opaque clouds.*

*We compared the albedo decrease in MAR simulations to the forcing temperatures in the ESM. Figure~/ref{ALTAS} reveals that MAR forced by CNRM-CM6-1 projects a stronger albedo decrease over the WAIS sector associated with large warming rates compared to MAR forced by CESM2. This results from the above discussed discrepancies in cloud properties leading to different melt rates and associated changes in albedo. While the more liquid precipitation in some MAR experiments could contribute to further decrease the surface albedo, a sensitivity experiment in MAR forced by CNRM-CM6-1 where rainfall amounts were set to 0 reveals no difference with the original MAR CNRM-CM6-1 experiment. It can be explained by the larger increase in melt compared to rain and then the predominant effect of melt increase on the albedo decrease. This suggests that differences in liquid precipitation created by clouds do not further strengthen melt differences, at least for the precipitation rates projected by our different MAR experiments.*

*Finally, our projections also illustrate the competitive effect of clouds as they can contribute to the surface albedo decrease through increasing LWD radiations but also reflect SWD and then protect the surface for another source of incoming energy. Their influence on absorbed SWD mainly depends on the surface albedo but also on the rate at which SWD is projected to decrease due to an increase in CC and/or COD \citep{bintanja1996}. In warmer climates (after 2100), clouds could be more reflective than the ice-covered surface, as summer surface albedo is projected to decrease. These warmer conditions could reverse the summer cloud radiative effect, reducing melt, similarly as over the dark ablation zone of the Greenland Ice Sheet \citep{hofer2017, wang2019}, suggesting a growing importance of surface albedo in determining the future cloud radiative effect but also in general SEB and melt changed over the AIS.*

4. Are the melt differences really due to clouds?

If I am not mistaken, the only forcing parameters from the ESMs that directly influence cloud phase and fractions are temperature and specific humidity. Hence, the results of this study mainly highlight the different equilibrium states of MAR given a set of forcing conditions, as you state in Section 3.2.1. Then, it is not completely clear to what extent the results are related to the actual cloud physics, or just to a different climate state that corresponds with a different cloud setup, including influences of other internal feedbacks you are not discussing. I would like to see some more discussion about this and the fact that the results you present are not due to differences in model physics, but due to differences in forcing conditions. What cloud differences do we see in the forcing ESMs themselves? Are they similar or does MAR really change the cloud behaviour?

All our experiments were done using the exact same MAR physics meaning the use of the same common set-ups for the cloud and radiative schemes in all the experiments. Since we only changed the boundary conditions (specific humidity, temperature, wind, pressure, SST

and SIC), all discussed differences (ie, melt, fluxes or cloud particle contents) results from different responses of MAR to the forcing ESMs. We indeed "highlight the different equilibrium states of MAR given a set of forcing conditions" that explained the differences in melt (ie more liquid clouds, more longwave, more melt). Since LWP tends to increase with higher temperature, we have tried to carry out most of the analyses at common warming rates to remove this influence (as we initially did for melt). For instance, we attributed the larger melt increase in MAR forced by CNRM-CM6-1 to more liquid clouds at the end of the century compared to MAR forced by CESM2. The different future clouds result from different "equilibrium states" (or physical response) in MAR themselves resulting from a combination of vertical temperature and humidity changes as explained L248-261, Fig5 and Table S2.

[Figure]

Fig R11: Humidity change (kg kg-1) at 850hPa projected by the ESMs over 2071-2100 compared to 1981-2010. The red box corresponds to the MAR domain.

To further illustrate the differences in humidity suggested by Fig. 5, we have compared the humidity change at 850 hPa (Fig. R11) and 500 hPa (Fig. R12) projected by the ESMs over 2071-2100 compared to 1981-2010. All the ESMs suggest an increase in humidity at both levels that is strongly determined by their warming (as per the Clausius-Clapeyron relation). NorESM1-M then suggests the lowest increase and CNRM-CM6-1 and CESM2 the highest ones. As already revealed by Fig. 5, CESM2 simulates a larger increase at 500 hPa than CNRM-CM6-1 (Fig Rx) while CNRM-CM6-1 simulates a larger increase at 850 hPa. These differences are relatively uniform around the boundaries of MAR, except near the Kerguelen Islands where CESM2 projects an increase in humidity at 850hPa higher than CNRM-CM6-1. We therefore think Fig. 5 is representative of humidity changes even if it shows average values and suggest adding these new maps in the Supplement to support the analysis and change L254:

*At the lateral boundaries, the CESM2 experiment reveals a future stronger increase in specific humidity above 2000~\unit{m asl} than the CNRM-CM6-1 one. The pattern is*

*opposite below 2000~\unit{m asl} where the future CNRM-CM6-1 atmosphere is characterised by stronger low-level humidity advection (Fig.~\ref{veticalqqp}a).*

*To*
*At the lateral boundaries, the CESM2 experiment reveals a future stronger increase in specific humidity above 2000~\unit{m asl} than the CNRM-CM6-1 one. The pattern is opposite below 2000~\unit{m asl} where the future CNRM-CM6-1 atmosphere is characterised by stronger low-level humidity advection (Fig.~\ref{veticalqqp}a).* **Supplementary maps (Fig. SX and Fig. SY) illustrate that these discrepancies are relatively spatially uniform around the boundaries of MAR.**

[Figure]

Fig R12 Humidity change (kg kg-1) at 500hPa projected by the ESMs over 2071-2100 compared to 1981-2010. The red box corresponds to the MAR domain.

Concerning the ESM clouds, unfortunately we did not download these ESM outputs from the ESGF nodes and the CESM2 version (and ensemble) we used is no longer available on the nodes (last access: 17/12/2021) so we cannot answer if the higher increase in liquid clouds are only simulated by MAR forced by CNRM-CM6-1, but we are confident that it should be similar in the ESM considering the patterns in humidity changes in the ESM themselves (Fig R11 and Fig R12).

5. In other studies such as King et al., 2015 or specific ESM intercomparison studies, differences in clouds are really due to differences in model physics which strengthen the case that the cloud representation in climate models should be improved, but that is not the case in this study. Your concluding remark: "…our study stresses the need to improve cloud representation in climate models to better constrain SLR projections" therefore does not really relate to the results of this study.

Our intention is to highlight that since clouds can lead to large uncertainties and melt spreads for a similar warming (while melt amounts are often linked with temperature changes (Trussel et al., 2015; Donat-Magnin et al., 2021; Kittel et al., 2021)), it is important to improve them to reduce the uncertainties. We suggest to remind here MAR biases in the representation of liquid clouds and change the conclusion accordingly.

*While MAR projections reveal significant melt differences using different ESM forcings \citep{kittel2021,gilbert2021}, we emphasise here that none of these projections is more plausible than any other and that the purpose of this study is, on the contrary, to highlight the physical factors that can lead to large uncertainties in Antarctic melt projections. The warming projected by the ESM forcing is the main factor controlling absolute melt differences, but we suggest that clouds and their phase are important factors contributing to the spread in melt and by extension surface mass balance projections of the AIS for the same warming rate. Furthermore, a recent study with MAR \citep{Letoumelin2020} has revealed significant changes in LWD due to drifting snow, a process not modelled in our study, suggesting that drifting snow could further contribute to the spread in melt projections. While climate models (including MAR) tend to poorly simulate clouds over the present \citep{Gallee2010, king2015, silber2019, gilbert2020, Mattingly2020, Mulmenstadt2021}, our study stresses the need to improve cloud representation in climate models to better constrain SLR projections.*

to

*While MAR projections reveal significant melt differences using different ESM forcings \citep{kittel2021,gilbert2021}, we emphasise here that none of these projections is more plausible than any other and that the purpose of this study is, on the contrary, to highlight the physical factors that can lead to large uncertainties in Antarctic melt projections. The warming projected by the ESM forcing is the main factor controlling absolute melt differences, but we suggest that clouds and their phase as simulated in MAR are important factors contributing to the spread in melt and by extension surface mass balance projections of the AIS for the same warming rate. Recent studies with MAR \citep{Letoumelin2020, Hofer2021} have revealed significant changes in LWD due to drifting snow, a process not modelled in our study, suggesting that drifting snow could further contribute to the spread in melt projections.* **Furthermore, MAR seems to underestimate the present summer LWP compared with CloudSat-Calipso estimates (not shown). Given the high sensitivity of the projected melt to these liquid clouds, future work should improve the cloud representation in MAR that could lead to revised melt projections over the Antarctic ice shelves.**

**Minor comments**

P1l10-12: What do you mean by "increasing melt differences"? Do liquid containing clouds increase the melt differences, or do you mean something else? Please rephrase.

We suggest to replace:

*By increasing melt differences over the ice shelves in the next decades, liquid-containing clouds could be a major source of uncertainties related to the future Antarctic contribution to sea level rise.*

*by*
*Since liquid-containing clouds are projected to increase the melt spread associated with a given warming rate, they could be a major source of uncertainties related to the future Antarctic contribution to sea level rise.*

P2l17: If you emphasise surface albedo, you should also note the effects of snowfall and fresh snow albedo due to differences in cloud(cover)
See our response to major comments #3.

P2l36: "Bright surfaces". Why does this matter? You do not come back to this
We suggest to remove and keep:
*Quantifying the influence of clouds on the SEB remains challenging over high latitudes.*

P2l36-44: This paragraph is not logically arranged. You note the lack of observations but you do not repeat this and why do high ECSs matter? Try to rephrase and shorten this paragraph.
This paragraph has been modified, see our response to major comments #1

P2l43: The term RCM should be introduced here, after you did introduce the ESMs. And how is an ESM not polar oriented, and a high-resolution model is polar oriented?
This paragraph has been modified, see our response to major comments #1. Concerning the "ESM not polar oriented and high-resolution model", this sentence was probably unclear and we thank the reviewer for the remark. It was removed when the paragraph has been rewritten.

P2L61-62: "correctly represents present Antarctic surface melt" -> "present-day" and this should be much shown in much more detail. Compensating errors might result in a correct modelling of surface melt production, but a more comprehensive study should be done. For instance: if the compensating errors are due to compensating clear-sky and cloudy conditions, your results will be hard to believe. Try to show that the contemporary contribution of cloud radiative feedback to the surface melt is accurately modelled.

We modified "present" as you suggest. Despite we think a more comprehensive study about MAR present-day melt is not the subject of this manuscript, we refer to our response to major comments #1 for the evaluation. To our knowledge, there is no way enabling the assessment of cloud radiative feedback to the surface melt, especially in the observations.

P3l63: "compensating errors" is too vague. Just leave this sentence as "This suggests a correct representation of the SEB, but…."
See #R2 's comment who approved this passage. We suggest to conciliate both comments and change accordingly:
*This suggests a satisfactory representation of the SEB through compensating turbulent fluxes and in general compensating errors whose impacts on the future SEB and melt is difficult to assess.*

P3L64-66: "the projected spread in melt" comes out of the blue; can you introduce this in more detail as it is the focus of this manuscript?

This is now better explained in the introduction, see our response to #R1.1

P5L126: You average over the ESMs as well? This is not completely clear.
Thanks for pointing out this unclear sentence. We suggest to replace:

*The reference (present) period for computing the anomalies in this study is taken as the summer (December-January-February, DJF) average from 1981 to 2010 for MAR over ice shelves (melt, SEB components, cloud amount and properties, surface albedo) and ESMs over the Antarctic region, i.e 90°S--60°S (near-surface warming).*

By

*The reference (present) period for computing the anomalies in this study is taken as the summer (December-January-February, DJF) average from 1981 to 2010 for MAR over ice shelves (melt, SEB components, cloud amount and properties, surface albedo)**. In the same way, we define the ESM warming as the mean changes in the summer (DJF) near-surface temperatures over the Antarctic region, i.e 90°S--60°S (near-surface warming) compared to 1981-2010**.*

P5L132: Why are you using cumulative numbers? Either convert them into SLR equivalents, or just use the climatological averages in mmWE or Gt…These very large numbers are hard to interpret **and** P5L138: Same as above but now with cumulative fluxes. How to interpret a cumulative flux of 443.7 W?
We use cumulative numbers as they highlight the total melt production over a century. Converting melt into SLR equivalents would be equivalent but physically incorrect as 1) thickness variations of the ice shelves do not contribute to SLR 2) surface melt water could refreeze and is then different from runoff. In the same way, we use cumulative fluxes as they represent the absorbed energy over the same period. We think it is more coherent, but depending on the Reviewers' and Editor's opinions, we could use averaged fluxes. Note that Fig1 will be modified accordingly our response #R1.2

P7L153: Some latex errors here?
Thanks, it is now corrected.

P7L156-159: Sensible heat can be a very local effect and might be smoothed out by your choice of presenting Antarctica wide averages. Your results might be dominated by the Antarctic Peninsula, which has the largest melt rates by far. It would be interesting to add a bit more regional studies to this study and grasp whether the LWN/SWN – melt correlation is consistent and homogenous across the continent.
We indeed found a strong contribution of sensible heat over the Peninsula in the averaged values. See our response to major comments #2.

P8L185-190: Why so many supplementary Figures referenced? If you need them in the main manuscript, use them! TC has no figure limit, and they can easily be added as an additional column in Fig.2 as well. (this argument holds for all other references of supplementary figures in the rest of the paper).
We think our supplement respects TC requirements and we prefer to not break our argumentation with technical details: "In no case can supplementary material contain

scientific interpretations or findings that would go beyond the contents of the manuscript."
"technical or theoretical developments that do not need to be included in the main text should be included as appendices". Most mentions related to Figures in the Supplement could have been "No shown" but we wanted to remove potential doubts (they are only called once and only support one sentence). We will however replace the improved version of Fig. S4 and S5 (see major comments #2) in the main manuscript. We will follow the Editor's opinion.

P10L217: Sentence is unclear. I do not see the value 0.7 reached and surely not by 2040-2060.
This is indeed a mistake, thanks for pointing it out. This paragraph has been removed according to our response to reviewer#2.

P12L260: What is the effect of rain? Should you add this to the melt flux? As this is very important for surface albedo. (and more..)
See our response to major comments #3.

P12L264: Again, please consider the effects of precip (snow and rain) on the surface albedo
See our response to major comments #3.

P12L268: Isn't this strong increase in CNRM-CR6 (not also) explained by the larger amounts of liquid precipitation?
See our response to major comments #3.

P13L274: Again, I think the influence of precipitation changes should not be passed
See our response to major comments #3.

**Figures:**
Figure 1: What is the colorbar for 1b, and what is the orange colouring we see?
We apologise for the missing information and thank the reviewer. The colorbar was the ice shelf fraction as defined in MAR. Figure 1b will be removed, see our response to major comments #2

Figure 2: What is happening with CC in CNRM from 2060 onwards?
The low-level humidity advection in this experiment favours the formation of low-level clouds with a high COD (ie, high liquid content and relatively warm) and high cloud cover (which are stratocumulus clouds at these latitudes). This increase is particularly noticeable on the large ice shelves (Ross and Ronne). See our response to major comments #2.

Figure 4-6: Isn't there a way to combine these plots (and make room for some of the supplementary figures to be included in the main?)
We will change Figure 4 and 6 according to our response to major comments #2

References:
Delanoë, J., Heymsfield, A. J., Protat, A., Bansemer, A., and Hogan, R.: Normalized particle size distribution for remote sensing application,Journal of Geophysical Research: Atmospheres, 119, 4204–4227, 2014.

Donat-Magnin, M., Jourdain, N. C., Kittel, C., Agosta, C., Amory, C., Gallée, H., Krinner, G., and Chekki, M.: Future surface mass balance and surface melt in the Amundsen sector of the West Antarctic Ice Sheet, The Cryosphere, 15, 571–593, https://doi.org/10.5194/tc-15-360 571-2021, 2021.

Gilbert, E., Orr, A., King, J. C., Renfrew, I., Lachlan-Cope, T., Field, P., and Boutle, I.: Summertime cloud phase strongly influences surface melting on the Larsen C ice shelf, Antarctica, Quarterly Journal of the Royal Meteorological Society, 146, 1575–1589, 2020.

Hofer, S., Amory, C., Kittel, C., Carlsen, T., Le Toumelin, L., and Storelvmo, T.: The contribution of drifting snow to cloud properties andthe atmospheric radiative budget over Antarctica, Geophysical Research Letters, 48, e2021GL094 967, 2021.

Inoue, J., Sato, K., Rinke, A., Cassano, J. J., Fettweis, X., Heinemann, G., Matthes, H., Orr, A., Phillips, T., Seefeldt, M., et al.: Clouds and radiation processes in regional climate models evaluated using observations over the ice-free Arctic Ocean, Journal of GeophysicalResearch: Atmospheres, 126, e2020JD033 904, 2021

Jakobs, C. L., Reijmer, C. H., Smeets, C. P., Trusel, L. D., Van De Berg, W. J., Van Den Broeke, M. R., and Van Wessem, J. M.: A benchmark dataset of in situ Antarctic surface melt rates and energy balance, Journal of Glaciology, 66, 291–302, 2020.

King, J., Gadian, A., Kirchgaessner, A., Kuipers Munneke, P., Lachlan-Cope, T., Orr, A., Reijmer, C., van den Broeke, M., Van Wessem, J., and Weeks, M.: Validation of the summertime surface energy budget of Larsen C Ice Shelf (Antarctica) as represented in three high resolution atmospheric models, Journal of Geophysical Research: Atmospheres, 120, 1335–1347, 2015.

Kittel, C.: Present and future sensitivity of the Antarctic surface mass balance to oceanic and atmospheric forcings: insights with the regional climate model MAR, Ph.D. thesis, University of Liège, Liège, http://hdl.handle.net/2268/258491, 2021.

Lenaerts, J. T., Van Tricht, K., Lhermitte, S., and L'Ecuyer, T. S.: Polar clouds and radiation in satellite observations, reanalyses, and climate models, Geophysical Research Letters, 44, 3355–3364, 2017.

Listowski, C., Delanoë, J., Kirchgaessner, A., Lachlan-Cope, T., and King, J.: Antarctic clouds, supercooled liquid water and mixed phase,investigated with DARDAR: geographical and seasonal variations, Atmospheric Chemistry and Physics, 19, 6771–6808, 2019.

Mattingly, K. S., Mote, T. L., Fettweis, X., Van As, D., Van Tricht, K., Lhermitte, S., Pettersen, C., and Fausto, R. S.: Strong summer atmospheric rivers trigger Greenland Ice Sheet melt through spatially varying surface energy balance and cloud regimes, Journal ofClimate, 33, 6809–6832, 2020

Palerme, C., Kay, J. E., Genthon, C., L'Ecuyer, T., Wood, N. B., and Claud, C.: How much snow falls on the Antarctic ice sheet?, The Cryosphere, 8, 1577–1587, https://doi.org/10.5194/tc-8-1577-2014, 2014.

Van Tricht, K., Lhermitte, S., Lenaerts, J. T., Gorodetskaya, I. V., L'Ecuyer, T. S., Noël, B., van den Broeke, M. R., Turner, D. D., and vanLipzig, N. P.: Clouds enhance Greenland ice sheet meltwater runoff, Nature communications, 7, 1–9, 2016.

Vignon, É., Alexander, S., DeMott, P., Sotiropoulou, G., Gerber, F., Hill, T., Marchand, R., Nenes, A., and Berne, A.: Challenging and improving the simulation of mid-level mixed-phase clouds over the high-latitude Southern Ocean, Journal of Geophysical Research:Atmospheres, 126, e2020JD033 490, 2021.

Wille, J. D., Favier, V., Gorodetskaya, I. V., Agosta, C., Kittel, C., Beeman, J. C., Jourdain, N. C., Lenaerts, J. T., and Codron, F.: Antarcticatmospheric river climatology and precipitation impacts, Journal of Geophysical Research: Atmospheres, 126, e2020JD033 788, 2021

---

## Author Comment (AC2)

**Reviewer #2 - Tri Datta**

We would like to thank the reviewer #2 for her interesting comments and suggestions which helped us to improve our manuscript. Please find our responses in blue below your comments. Italic text represents unchanged text from our manuscript while bold text highlights our suggested changes.

This manuscript presents an analysis of the importance of cloud properties in driving surface melt over Antarctic ice shelves in the future (to 2100) , comparing these to a 1981-2010 reference period. This uses the MAR model forced at the boundaries with 4 ESMs (ACCESS1.3, NorESM-1-M, CRNM-CM6-1 and CESM2) in the RCP8.5 (for CMIP5 models) and SSP585 (for CMIP6 models). The authors examine potential drivers for surface melt beginning with energy balance components, identify the importance of clouds, and present a strong analysis of properties which contribute most to differences in melt produced by each ESM-forced-version of MAR. I commend the authors on a very well organised argument and believe that this will eventually be a strong contribution to the understanding of future surface melt in Antarctica, although several important aspects are currently missing, which could be addressed with additional figures and analysis.

**Specific Comments**:
1) The integration of all ice shelves may be hiding processes which vary spatially

As an example, the authors specifically admit that the SEB is impacted by SHF values only in certain regions. We note that one such region is the Larsen C ice shelf, where a substantial amount of total surface melt occurs. At the 35 km spatial resolution, surface melt would necessarily be poorly-represented over the Larsen C ice shelf in this version of MAR. A more meaningful analysis (making this manuscript an excellent companion to Gilbert and Kittel, 2021) would be to essentially conduct the organisation of this study, but with ice shelves divided regionally.
We acknowledge a regional analysis could improve the manuscript, so that we will include it. See our response to #R1.

For plots (i.e. Figures 2,4, 6, S1, S2) these would benefit from a map showing differences ( as in Fig. S3). We note that on line 164, the authors mention the thickening of the future planetary boundary layer over ice shelves of West Antarctica – it would be relevant to show whether this was demonstrated in East Antarctica as well independently.
Ok for the plots Figs.4,6, S1 and S3 (note that the main interesting part of this plot is the SHF anomaly that is already represented spatially in Fig. S4). We will also add maps of CC and COD changes. (also see our response to #R1 for the regional analysis)

Unfortunately we did not save the height of the PBL in our simulations and the variables at our disposal do not enable us to recalculate it afterwards either. We will ensure that PBL is added to the default outputs of the model.

Additionally, it would be beneficial to see similar maps of averaged values for forcing fields (in Supplemental Figures) to illustrate the spatial characteristic of the differences in forcing. By integrating, we have no picture on the spatial characteristics which are driving this (i.e.

are the differences in moisture at lower altitudes vs higher altitudes dominant in West Antarctica but not East Antarctica)
See our response to #R1. We will add those maps in Supplement.

2) A more rigorous account of changes in albedo

The differences in albedo are mentioned briefly, but this seems to be a major driver in the overall differences shown, and there is no discussion about how this is impacted by snowfall events. While I think that a thorough examination of precipitation trends is outside the scope of this manuscript, an analysis of albedo differences (in a map) as well as snowfall differences (in a map) would strengthen the manuscript significantly.
We also think that a thorough examination of precipitation is outside the scope of this manuscript, especially since it was already presented in Kittel et al., 2021. Snowfall and rainfall differences are discussed in their section 4.1.2 (including maps and tables). However, we acknowledge that a map with albedo differences will contribute to improve our manuscript and will therefore be added (as well as a map with rainfall changes as it was not included in Kittel et al., 2021).

About the role of snowfall on albedo, it was not originally discussed as Shortwave Net contribution is relatively close for a same warming rate suggesting that the effect of the surface albedo decrease is compensated by the "parasol" effect of clouds (also stronger in the experiment where the surface albedo decrease the most, ie MAR forced by CNRM-CM6-1). The regional analysis sheds light on discrepancies in the SWN contributions so that we will discuss the albedo differences where it matters. See our response to #R1 where we also discuss the influence of rainfall on the surface albedo.

3) A greater discussion of biases in cloud properties that are present in historical runs of MAR

To my knowledge, none of the evaluations of MAR present a comparison of biases in cloud properties (as compared to observations, i.e. CALIPSO). If I've missed something, a reference and a short discussion would be relevant. If not, then some level of validation of MAR's representation of cloud properties in the historical record would be directly germaine to this study.
See our response to #R1 (and suggested modifications of our manuscript accordingly).

**Technical Corrections:**

L 42: I could not find the reference to lower future melt changes in ESMs in Kittel et al., 2021. Could the authors clarify (identify a figure/section)?
This is notably highlighted in their Figure S13b where MAR simulations always suggest a stronger decrease in SMB than ESM simulations. This is certainly an extrapolation but that seems reasonable since SMB over the ice shelves will be dominated by runoff (due to melt) changes. We will however remove the sentence as we suggest to modify the introduction to better emphasise the aim of our study (understand why there are large melt differences even of the same warming rate). See our answer to #R1.

L 61: Use of the word "correct" twice in proximity. Additionally. I would suggest, "presents well" as opposed to "correctly". Additionally, Kittel et al., 2021 refers to future runs, rather than an evaluation of a historical run. Perhaps referencing Agosta et al., 2019 would be more accurate.

An evaluation of MARv3.11 forced by ERA5 against weather station data is well available in Kittel et al. (2021) (while this kind of evaluation was not done in Agosta et al., 2019), see Section 2.1.2 and Supplementary Section S1 in Kittel et al. (2021). See also comment to reviewer 1 where we modified the sentence (avoiding using correct twice in proximity, thanks).

L 64: "difficult to assess". This is a good way to declare this complexity without making dishonest claims. Thanks for that.

Thank you for noticing.

L 110: Make clearer exactly how these melt projections used climate models.

We suggest to change:

*Most of the projections of the Antarctic melt have been performed in the frame of the 5th phase of the Coupled Intercomparison Project (CMIP5) \citep[e.g.,][]{Trusel2015}, while more recent climate models from CMIP6 now project stronger warmings at both local (Antarctic) and global scales.*

to

*Most of the projections of the Antarctic melt **have been based on direct outputs of models \citep[e.g.,][]{Seroussi2020}** from the 5th phase of the Coupled Intercomparison Project (CMIP5), **or derived from them using statistical regression \citep[e.g.,][]{Trusel2015}**, while more recent climate models from CMIP6 now project stronger warmings at both local (Antarctic) and global scale*s.

L 171: Awkward sentence. Suggestion: "This could be explained by accounting for the ECS capturing the greater warming over the Antarctic region simulated by CNRM-CM6-1 (+8.5°C vs 7.7°C for CESM2 in 2100 compared to 1981-2010).

Thanks for the suggestion, we will change accordingly.

L 173: replace "this ESM" with "CRNM-CM6-1" for clarity

Changed, thanks.

L 176: relatively – (remove dash)

Corrected

L 182: "suggests a low influence on LWD". Clarify the discrepancy by comparing the quantity

The sentence was unclear, there is no difference in LWD due to GHG as it can be considered as an supplementary effect that is equivalent between our simulations. We propose to remove this part of the sentence.

L 206: "southern" == "austral"
Changed.

L 210: Why are clear and cloud sky conditions treated separately? Could you clarify the reason or separate the analysis accordingly?
They are not treated separately. We mean that the COD as computed in MAR could then increase for three reasons: optically thicker clouds (but same occurrence), more clouds (but unchanged cloud composition), or a combination of both. Figure 2 indeed reveals a COD increase in MAR forced by CNRM-CM6-1 as CC increases, but it also shows that a same increase in CC in MAR forced by CESM2 leads to a higher increase in COD.

L 220: Could you demonstrate the saturation of LWD for large COD increases in supplemental?
Following your request, we extended the relations expressed in Fig. 3 for large COD increases. Since the relation is expressed with an exponential function, we computed the values given L220 by assuming a minimal variation of LWD for a COD change using a very arbitrary way. We tried to improve the computation by fixing the variation of LWD threshold to 0.1% associated with a change in the COD anomaly of 0.01. This gave a correct detection of the COD anomaly associated with negligible change in LWD (ie, where LWD anomaly could be considered to stop increasing) for at least the CESM2 and NorESM1-M experiments (Fig R13), and a satisfactory one for the other experiments.

[Figure]

Fig. R13: Relation between LWD summer anomalies and COD summer anomalies. Summer longwave downwelling radiation (W m−2 ) versus mean cloud optical depth anomalies during summer (-) projected by MAR driven by ACCESS1.3 (a), CESM2 (b), CNRM-CM6- 1 (c), and NorESM1-M (d) compared to the summer reference period (1981–2010). The exponential regression as well as corresponding determination coefficient ($R^2$ , p «0.01) is indicated for each experiment. A 5-year running mean has been applied on the anomalies. The vertical dashed line represents the COD anomaly where LWD anomaly could be considered to stop increasing.

However, these values (+1.07 for ACCESS1.3, +0.63 for CESM2, +1.19 for CNRM-CM6-1, and +0.49 for NorESM1-M) are based on the strong extrapolation of an exponential function that is likely very uncertain (difficult to fit) and still using an arbitrary methodology. Given all of that, we suggest to replace L221-L226 :

*We extrapolate our projections based on equations from Fig. 3, to find that increase in LWD associated to an increase in COD would stop when COD equals 1.22 (+0.96 compared to present values) (ACCESS1.3), 1.10 (+0.96) (NorESM1-M), 1.78 (+0.91) (CNRM-CM6-1), 1.2 (+0.89) (CESM2). Since these values are not reached before 2100 in our simulations, the future LWD increase is supposed to remain sensitive to cloud optical properties during the whole 21st century, including for high warming rates as projected by CNRM-CM6-1 and CESM2.*

by

***The relations expressed in Fig. 3 suggest that the sensitivity of the LWD increase would progressively stop for (very) large increases in COD.*** *As these values are not reached before 2100 in our simulations, the future LWD increase is supposed to remain sensitive to cloud optical properties during the whole 21st century, including for high warming rates as projected by CNRM-CM6-1 and CESM2.*

References:

Agosta, C., Amory, C., Kittel, C., Orsi, A., Favier, V., Gallée, H., van den Broeke, M. R., Lenaerts, J., van Wessem, J. M., van de Berg, W. J., et al.: Estimation of the Antarctic surface mass balance using the regional climate model MAR (1979-2015) and identification of dominant processes, The Cryosphere, 13, 281–296, 2019.

Kittel, C., Amory, C., Agosta, C., Jourdain, N. C., Hofer, S., Delhasse, A., Doutreloup, S., Huot, P.-V., Lang, C., Fichefet, T., and Fettweis, X.: Diverging future surface mass balance between the Antarctic ice shelves and grounded ice sheet, The Cryosphere, 15, 1215–1236, https://doi.org/10.5194/tc-15-1215-2021, 2021.

---

## Author Comment (AC3)

**Reviewer #3**

The authors provide an overview of the driving forces of surface melt over the ice shelves over Antarctica. Although the work is mainly confirming the results of other studies, it is one of the first to provide a quantitative assessment of these driving forces towards the future, which is a step forward in the scientific understanding of how models interact with the surface.

We would like to thank the reviewer #3 for reading the manuscript and the interesting comments which helped us to improve our manuscript. Please find our responses in blue below your comments. Italic text represents unchanged text from our manuscript while bold text highlights our suggested changes.

**Main comments:**

- The paper is well structured & contains a lot of interesting information regarding the representation of melt in future simulations. One aspect that is not discussed however is the effect of precipitation on surface melt. A higher number of clouds would possibly also result in higher precipitation numbers, which would increase the surface albedo. This counteracts part of the warming induced by the increase in liquid clouds and LWD. It would be nice to see the contribution of precipitation and surface albedo in the results.
See our response to #R1 that includes theoretical approaches and sensitivity experiments to discuss the influence of the amount of snow and rainfall compared to melt.

- The paper mainly discusses average changes towards the future. However, most of the large melting occurs during 'events' nowadays. With the increase in general surface temperatures towards the future, I am wondering if these individual melt events become of lesser / higher importance & that individual events will still be the driver of most melt or that temperatures will increase to such a level that melt will occur during the whole summer.

Since we mostly aim to discuss the physical reasons behind melt differences over the century, we indeed focus our analyses on average changes (or climate changes). However, we acknowledge that changes in the influence of individual events could also contribute to differences and then improve our analysis. We compared the amount of mean melt produced during the strongest events (daily melt above the p95 of the climate period) to the mean total amount of melt for the present period (1981-2010) and the future period (2071-2100).

$$Ratio = \sum_{dailymelt>P95}^{summer} \frac{melt}{\sum\limits_{allday}^{summer} melt}$$

As expressed by Eq. R1, high values (~1) of the ratio suggest that total summer surface melt is mainly produced during strong events, while lower values highlight the contribution of more numerous "smaller" events in the total production. As mentioned by #R3, the ratio is close to 1 (and generally over 0.8) over most of the Antarctic ice shelves over the present period, the peninsula excepted (Fig. R14). This underlines that in the current climate over most of the ice shelves, melt is associated with events (e.g, Atmospheric Rivers or Foehn).

On the contrary, our experiments all suggest a decrease in the contribution of strong events to the total summer melt in the future and that melt could occur during more days in the whole summer (Fig. R15.) This would also have consequences in terms of modelling: properties during very isolated events (AR or at least maritime intrusions) while it could be different for future climates.

We suggest to add in S3.1:

*The contribution of a few specific events to produce melt over the Antarctic ice shelves is projected to change. We compared the amount of mean melt produced during the strongest summer events (daily melt above the p95 of the climate period) to the mean total summer amount of melt for the present period (1981-2010) and the future period (2071-2100). Over present-day conditions (Fig.~S5), this ratio is high (higher than 80\%, the peninsula excepted) suggesting that melt mainly occurs during specific events such as atmospheric rivers \citep{Wille2019}. On the contrary, all the MAR experiments project a much lower contribution of these specific events in the total summer melt (Fig.~S6). This suggests that melt will occur during more days in the whole summer.*

[Figure]

Fig R14: Mean ratio of the melt produced during melt days stronger than P95 to the total surface melt in summer over 1981-2010 as simulated by MAR forced by ACCESS1.3 (a), CESM2 (b), CNRM-CM6-1 (c), NorESM1-M(d).

[Figure]

Fig R15: Mean ratio of the melt produced during melt days stronger than P95 to the total surface melt in summer over 2071-2100 as simulated by MAR forced by ACCESS1.3 (a), CESM2 (b), CNRM-CM6-1 (c), NorESM1-M(d).

The paper is clear and well written, i only have a few **specific comments**:

- When first reading the title of the paper, I immediately thought of the paper of Van Tricht (https://doi.org/10.1038/ncomms10266). Despite dealing about a similar subject (although another ice sheet), I think a reference to this work is valid somewhere in the introduction. A same set of techniques is also used in the methodology & results section and in the discussion, one could relate to the differences between the Greenland & Antarctic Ice Sheet
We added a reference to Van Tricht et al (2016) in the introduction:

*The net cloud radiative effect - the balance between these opposite contributions - is notably determined by the surface albedo \citep{bintanja1996,hofer2017}, and cloud properties, i.e their temperature \citep{stephens1984}, structure \citep{barrett2017, gilbert2020}, and water phase (ice or liquid) \citep{lachlan2010,**van2tricht2016**,hines2019,gilbert2020}.*

and

Clouds currently warm the Antarctic Ice Sheet (AIS) surface \citep{pavolonis2003,van2006}. While the highly-reflective snow already prevents significant absorption of solar downwelling radiation (SWD) in summer, clouds act as another source of incoming energy in the infrared spectrum, which can heat and melt snow \citep{bintanja1996,van2006} ***similarly as over bright surfaces of the Greenland Ice Sheet \citep{van2tricht2016}.***

- Specify on line 134 that NorEsm is the lowest range model & CNRM is the upper range (instead of line 140)

Change as requested:

from

*MAR driven by NorESM1-M simulates a cumulated melt increase of $\sim$8000 \unit{Gt} during the 21st century, while the increase reaches $\sim$31400 \unit{Gt} when MAR is driven by CNRM-CM6-1.*

to

*MAR driven by NorESM1-M simulates a cumulated melt increase of $\sim$8000 \unit{Gt} during the 21st century **(i.e the lowest melt increase)**, while the increase reaches $\sim$31400 \unit{Gt} when MAR is driven by CNRM-CM6-1 **(i.e the highest melt projection).***

---

## Author Response (AR2)

Dear Editor,

please find our revised manuscript as well as the point-by-point response to the reviewer's comments.

Minor changes in the manuscript are :

1) Addition of an outline of the sections as requested by the reviewer

2) Removal of Lines 183-189 (and associated section in the supplement) about the influence of specific events. This analyses interested a reviewer from the first round, but we agree with the reviewer here that is comes out of the blue and we don't think it is directly linked to the story of this study.

3) Corrections of small typo-mistakes.

Best regards,

C. Kittel on behalf of all co-authors

Review of Kittel

The authors conducted one of the most intensive revisions that I have seen, which I commend. They addressed all of my and the other reviewers concerns in large detail, and the entire manuscript has benefited as a result. Therefore I will recommend publication, after some of my small remaining comments below are addressed.

We would like to thank again the reviewer for the first comments we received as well for the new reading of our revised manuscript. This has really helped us to improve our manuscript and we are glad that we addressed them all.

Minor comments:
Introduction:
Especially in the new version of the manuscript, I miss an outline of the sections in the last paragraph. I.e.:" section 2 we discuss … ", "in section 3 ..". etc.
Added, thanks for the suggestion.

Lines 140-141: You note "a summer melt increase" but the figure just shows the cumulative anomaly.
Changed for: Our four simulations project an increase in cumulative summer melt over the ice shelves that strongly differs depending on the forcing ESM during the 21st century.

Lines 183-189: Where does this part come from? It comes out of the blue, and I am not in favor of discussing results solely based on figures in the Supplementary material.
This analyses interested a reviewer from the first round, but we agree that is comes out of the blue and we don't think it is directly linked to the story of this study. It is now removed but the results will remain available in the interactive discussion for people who might be concerned.

Figure 4, last line: "used"◊"using".
Thanks.

Throughout the edited parts (in blue in the track changes file), some new sentences were a bit hurried with minor typos. I will not note them all separately , but please check the final version well on small errors.
We checked it again and hope we found them all, thanks for the information.

---

## Author Response (AR3)

Dear Editor,

We would like to thank you for accepting our manuscript and for your support throughout the review process.

As wrote to the Editorial office, I tested the figures, and mainly the first one on the mentioned site. As far as the current colours are concerned, except for the strongest deficiencies (mono chroma), the figures are still readable. I think it is really good to be able to give everyone the opportunity to read the figures and that is why I had corrected the use of colours to improve the reading in my first submission. However, I don't think I could revise the figures any further as the use of colours would be too limited.

Best regards,
C. Kittel on behalf of all co-authors